



# Radar Imaging with EISCAT 3D

Johann Stamm[1], Juha Vierinen[1], Juan M. Urco[2], Björn Gustavsson[1], and Jorge L. Chau[2]

[1]Institute for physics and technology, University of Tromsø, Tromsø, Norway.
[2]Leibniz Institute of Atmospheric Physics, University of Rostock, Kühlungsborn, Germany

*Correspondence to:* Johann Stamm (johann.i.stamm@uit.no)

**Abstract.** A new incoherent scatter radar called EISCAT 3D is being constructed in Northern Scandinavia. It will have the capability of producing volumetric images of ionospheric plasma parameters using aperture synthesis radar imaging. This study uses the current design of EISCAT 3D to explore the theoretical radar imaging performance and compares numerical techniques that could be used in practice. Of all imaging algorithms surveyed, the singular value decomposition with regularization gave

the best results and was also found to be the most computationally efficient. The estimated imaging performance indicates that the radar will be capable of detecting features down to approximately 90x90 m at a height of 100 km, which corresponds to a $\sim 0.05°$ angular resolution. The temporal resolution is dependent on the signal-to-noise ratio and range resolution. The signal-to-noise ratio calculations indicate that high resolution imaging of auroral precipitation is feasible. For example, with a range resolution of 1500 m, a time resolution of 10 seconds, and an electron density of $2 \cdot 10^{11} \mathrm{m}^{-3}$, the correlation function

estimates for radar scatter from the E-region can be measured with an uncertainty of 5 %. At a time resolution of 10 s and an image resolution of 90x90 m, the relative estimation error standard deviation of the image intensity is 10 %. Dividing the transmitting array into multiple independent transmitters to get at multiple-input-multiple-output (MIMO) interferometer system is also studied and this technique is found to increase imaging performance through improved visibility coverage. However, an estimate shows that this reduces the signal-to-noise ratio. MIMO is therefore only useful for the most brightest

targets, such as meteors, polar mesospheric summer and winter echoes, and satellites. The results show that radar imaging of is feasible with the EISCAT 3D radar, and that the use of the MIMO technique should be explored further.

## 1 Introduction

One of the measurement challenges in the study of the Earth's ionized upper atmosphere using incoherent scatter radars (ISR)

is that the measurements often do not match the intrinsic horizontal resolution of the physical phenomena that is being studied. Conventional ISR measurements are ultimately limited in the transverse beam axis direction by the beam width of the radar antenna, which is determined by the diffraction pattern of the antenna. Even for large antennas, the beam width is typically around 1°. The mismatch between geophysical feature scales and horizontal resolution obtained by a typical ISR antenna is demonstrated in Figure 1, which shows an image of auroral airglow taken in the magnetic field aligned direction. Overlayed on



the image are the antenna beam diameters of three incoherent scatter radar antennas: EISCAT UHF, EISCAT 3D, and Arecibo. It is clear that the auroral precipitation has appreciable structure on scales smaller than the beam size. A conventional ISR measurement in this case will provide plasma parameters that are averaged over the area of the radar beam, preventing the observation of small sub-beamwidth scale structure. Only a radar with an antenna size of the Arecibo Observatory dish (305 m) would provide an antenna beam width that approaches the scale size of auroral precipitation.

Another measurement challenge for ISRs is temporal sampling of the spatial region of interest. Single dish radar systems can only measure in one direction at any given time, and the ability to move the beam into another direction depends on the speed at which the antenna can be steered. In addition comes the minimum integration time required to measure one position. It takes a long time to sample a large horizontal region, and even then, the measurements of different horizontal positions are obtained at different times.

In order to increase the spatial resolution of radio measurements without resorting to constructing an extremely large continuous antenna structure, a technique called aperture synthesis imaging can be used (e.g., Junklewitz et al., 2016). It relies on a sparse array of antennas to estimate a radio image with horizontal resolution equivalent to that of a large antenna. The correlation between the received signals can be used to produce an image of the brightness distribution of the radio source. This technique is widely used in radio astronomy to image the intensity of radio waves originating from different sky positions.

The application of aperture synthesis imaging for radar, i.e., aperture synthesis radar imaging (ASRI), has been used in space physics for observing high signal to noise ratio targets (Hysell et al., 2009; Chau et al., 2019). The currently available horizontal resolution is around $0.5°$ with Jicamarca, but down to $0.1°$ for strong backscatter (Hysell and Chau, 2012) in the case of field-aligned ionospheric irregularities; and $0.6°$ with the Middle atmosphere ALOMAR radar system (MAARSY) for polar mesospheric summer echoes (PMSE) (Urco et al., 2019).

In radar imaging, the measurements are in the so called visibility domain. Ensemble averages of the cross-correlation of complex voltages between two antennas represents a single sample of the visibility (Woodman, 1997; Urco et al., 2018). Throughout this article we will use "far field" for the region further away than the Fraunhofer limit of the radar. The region closer than the Fraunhofer limit we will call "near field". If the radar target is in the far field of the radar, the visibility domain is related via a Fourier transform to the horizontal brightness distribution or the radio image. Then, the measurements are samples of the Fourier transform of the spatial variation of backscatter strength, or brightness distribution, of the target (Woodman, 1997). The measurements are used to calculate the brightness distribution, or image, of the target.

So far, most of the incoherent scatter radar imaging has been done with a single transmitter and multiple receivers, thereby using a single-input multiple-output (SIMO) system. The number of measurements and degrees of freedom is here determined by the number of receivers and their relative locations. Instead of using only one transmitter, multiple transmitters can be used when performing radar imaging. This allows increasing the number of visibilities that can be measured, which can result in improved imaging performance, as long as the signal-to-noise ratio is sufficiently high. This technique is called multiple-input multiple-output (MIMO) radar (Fishler et al., 2006). The MIMO technique for increasing spatial resolution has recently been demonstrated with the Jicamarca radar when imaging equatorial electrojet echoes (Urco et al., 2018) and also with the





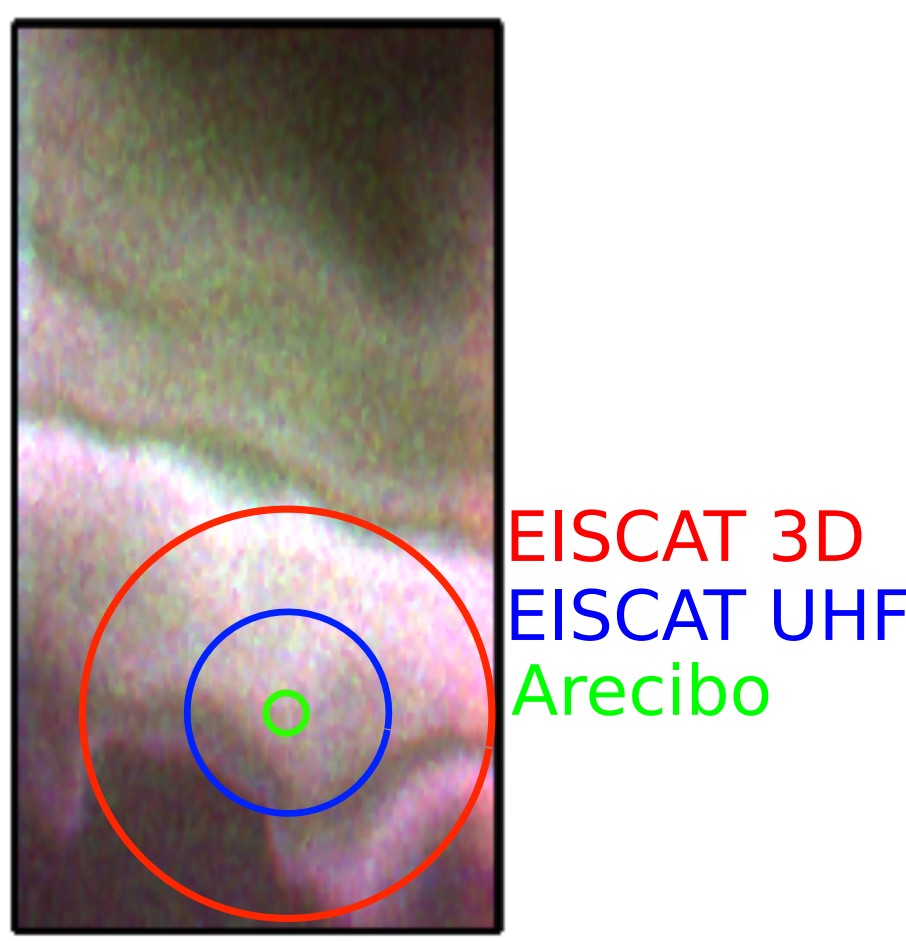

**Figure 1.** An image of auroral optical emission in the magnetic field aligned direction showing the horizontal distribution of auroral precipitating electron flux. Overlayed on top of the image are the beam widths of the EISCAT UHF, Arecibo, and EISCAT 3D radars, with approximately $0.5°$, $0.16°$, and $1.0°$ beam widths. Image from the Auroral Structure and Kinetics (ASK) instrument (Ashrafi, 2007), courtesy of D.K. Whiter.



MAARSY radar for imaging PMSE (Urco et al., 2019). The primary technical challenge with MIMO radar separating scattering corresponding to multiple transmitters on receive.

EISCAT 3D, from hereon referred to as E3D, is a new multi-static incoherent scatter radar that is being built in Norway, Sweden, and Finland (McCrea et al., 2015; Kero et al., 2019). The core transmit and receive antenna array will be located

in Skibotn, Norway (69.340° N, 20.313° E). There will be additional bi-static receiver antenna sites in Kaiseniemi, Sweden (68.267° N, 19.448° E) and Karesuvanto, Finland (68.463° N, 22.458° E). The core array of E3D will consist of 109 subarrays, each containing 91 antennas. The one-way half power full beamwidth (HPBW) or illuminated angle of the core array will be 1°. On transmission, the array is capable of transmitting up to 5 MW of peak power at a frequency of 233 MHz. Additionally, there are 10 receive-only outrigger antennas around the core array, providing longer antenna spacings that can be used for high

resolution ASRI. Imaging will already be necessary to maintain the perpendicular resolution constant in the transition from EISCAT VHF and UHF to E3D. It is possible that the EISCAT 3D radar can also be configured as a MIMO system, where the core array is separated into smaller subarrays, which act as independent transmitters at slightly different locations. During the design phase, Lehtinen (2014) investigated the imaging performance of possible layouts of E3D in the far-field. The study however does not include the current layout that is being built.

EISCAT 3D will not be able to measure radar echoes from magnetic field aligned irregularities, so it will not be possible to assume that the scattering originates from a two dimensional plane where the radar scattering wave vector is perpendicular to the magnetic field. All radar imaging will need to be done in 3D and mostly for incoherent scatter. This poses several two main challenges: 1) the signal-to-noise ratio will in typical cases be determined by incoherent scatter, which is much smaller than that used conventionally for ASRI; 2) there are more unknowns that need to be estimated, as at each range there is a 2D

image instead of a 1D image that needs to be estimated. While there is a good amount of literature on ASRI techniques in two dimensions (range and one transverse beam axis direction) for imaging field aligned irregularities (e.g., Hysell and Chau, 2012, and references therein), not much literature exists for imaging atmospheric and ionospheric features in three dimensions. An exception is Urco et al. (2019), who applied 3D imaging to observations of PMSE with MAARSY; and Schlatter et al. (2015), who used the EISCAT Aperture Synthesis Imaging array and the EISCAT Svalbard radar to image the horizontal structure of

Naturally Enhanced Ion Acoustic Lines (NEIALs).

For E3D, the Fraunhofer limit is at $2D^2/\lambda \approx 2000$ km, where $D \approx 1.2$ km is the longest baseline and $\lambda = 1.3$ m is the wavelength of the radar. Measurements of the ionosphere are therefore taken in the nearfield of the radar. Woodman (1997) describes a technique to correct for the curvature in the backscattered field with an analogy of lens focusing. In this study, a different approach has been taken, where the nearfield geometry is directly included in the forward model of the linear inverse

problems formalism. In this case, it is not possible to resort to frequency domain methods to diagonalize the forward model. This comes at an increase in computational complexity, but is not prohibitive in terms of computational cost with modern computers.

In this study, we will simulate radar imaging measurement capabilities of the upcoming EISCAT 3D radar. The study is divided into the following sections. In Sect. 2, we investigate the achievable time and range resolution of E3D, and how

they are connected. An expression for the cross-correlations between the received signals, taking into account the nearfield





geometry, is derived in Sect. 3. Sect. 4 describes the nearfield forward model for radar imaging and describes several numerical techniques for solving the linear inverse problem. This section also includes a study of imaging resolution based on simulated imaging measurements.

## 2 Time resolution

In this section, we will calculate the required integration time for a certain range resolution with E3D. The elementary radar imaging measurement is an estimate of the cross-correlation of the scattered complex voltage measured by two antenna modules. The integration time in this case is the minimum amount of time that is needed to obtain a error standard deviation for the cross-correlation estimate that is equal to a predefined limit. The estimation error of the cross-correlation determines the measurement error for the imaging inverse problem. By investigating the variance of the cross-correlation estimate using statistical properties of the incoherent scatter signal, we can decouple the problem of time and range resolution from imaging resolution,

allowing us to study the performance of the imaging algorithm with a certain measurement error standard deviation.

Our signal-to-noise calculations will be based on an observation of incoherent scatter from ionospheric plasma, which is the case with the smallest expected signal-to-noise ratio. We have ignored self-clutter, as the combination of the E3D core transmitter illuminating the target and a single 91 element receiver module will inevitably be within a low signal-to-noise ratio

regime, dominated by receiver noise.

We will first deduce an expression for the measurement rate, that is, how many measurements are taken per second. There are two factors that determine the maximum rate at which independent observations of the scattering from the ionosphere can be made: 1) The minimum inter-pulse period length, which we set to $d/\tau_p$, with $d$ the duty-cycle and $\tau_p$ the pulse length. 2) The incoherent scatter decorrelation time, which is inversely proportional to the bandwidth of the incoherent scatter radar spectrum

$B$. The maximum of these two time scales determines the frequency of independent measurements that can be made:

$$F_\mathrm{m} = \min\left(d\tau_\mathrm{p}^{-1}, B\right) \tag{1}$$

If a transmitted longpulse is divided into $N_P$ bits, the number of measurements per longpulse can be multiplied by $N_P$. If we additionally can assume that the autocorrelation function is constant, the number of lagged product measurements per transmit pulse is $N_P(N_P-1)/2$ because we also can use measurements with different time lags. For sake of simplicity, we assume that

all lags within a radar transmit pulse are equally informative. This is approximately the case for E-region plasma measured using E3D. The number of measurements per second is then

$$F_\mathrm{c} = F_\mathrm{m} N_\mathrm{P}(N_\mathrm{P}-1)/2. \tag{2}$$

Next, we will estimate the number of measurements needed for reducing the measurement error of an average cross-correlation measurement to a certain level. We consider a measurement model where a measurement $m$ is described by a

linear combination of the parameter we want to estimate $m = x + \xi$, where $x$ and $\xi$ are considered as proper complex Gaussian random variables with zero mean and variance of respectively $P_\mathrm{S}$ and $P_\mathrm{N}$. The noise power estimate $P_\mathrm{N}$ is assumed to have no





error. We estimate the signal power with

$$\hat{P} = \sum_{i=1}^{K} \frac{m_i \bar{m}_i}{K} - P_{\mathrm{N}}, \tag{3}$$

where the bar denotes complex conjugation. It can then be shown that

$$\mathrm{Var}(\hat{P}) = (\varepsilon P_{\mathrm{S}})^2 = \frac{(P_{\mathrm{S}} + P_{\mathrm{N}})^2}{K}, \tag{4}$$

5  where $\varepsilon$ is the relative standard deviation and K is the number of measurements (Farley, 1969). If we require the correlation function to have a relative uncertainty under a certain level $\epsilon$, f.ex. $\epsilon = 0.05 = 5\%$, the equation can be solved for $K$ in order to get the number of needed samples

$$K = \frac{(P_S + P_N)^2}{(\epsilon P_S)^2} = \left( \frac{\mathrm{SNR} + 1}{\varepsilon \cdot \mathrm{SNR}} \right)^2, \tag{5}$$

where SNR is the signal-to-noise ratio SNR. The integration time required to obtain a measurement with a certain level of

10  uncertainty is now

$$T = \frac{K}{F_c} = \frac{(P_S + P_N)^2}{(\epsilon P_S)^2} \frac{2}{F_{\mathrm{m}} N_{\mathrm{P}}(N_{\mathrm{P}} - 1)}, \tag{6}$$

or written as a function of SNR

$$T = \left( \frac{\mathrm{SNR} + 1}{\varepsilon \cdot \mathrm{SNR}} \right)^2 \frac{2}{F_{\mathrm{m}} N_{\mathrm{P}}(N_{\mathrm{P}} - 1)}. \tag{7}$$

The received signal power $P_{\mathrm{S}}$ can be found by the radar equation

$$P_{\mathrm{S}} = \frac{P_{\mathrm{tx}} G_{\mathrm{tx}} G_{\mathrm{rx}} \lambda^2 \sigma}{(4\pi)^3 R_{\mathrm{tx}}^2 R_{\mathrm{rx}}^2}, \tag{8}$$

where $P_{\mathrm{tx}}$ is the transmitted power, $G_{\mathrm{tx}}$ is the transmit and $G_{\mathrm{rx}}$ the receive gain, $\lambda$ is the radar wavelength, $\sigma$ is the scattering cross-section and $R_{\mathrm{tx}}$ and $R_{\mathrm{rx}}$ are the distance from the scattering volume to the transmitter and receiver. Assuming that the Debye length is much smaller than the radar wavelength, the effective scattering cross-section for a single electron in plasma (Beynon and Williams, 1978) is

$$\sigma_p = \sigma_e \left( 1 + T_e/T_i \right)^{-1}. \tag{9}$$

Here, $\sigma_e$ is the Thomson scattering cross-section $\sigma_e = 4\pi r_e^2$, $T_i$ is the ion and $T_e$ the electron temperature.

The total scattering cross-section can be found by adding up the cross-sections of all electrons in the illuminated volume $N_e V$,

$$\sigma = V N_e \sigma_p. \tag{10}$$

25  The scattering volume can be approximated using a conic section:

$$V = \frac{\pi \Delta r \tan^2\left(\frac{\theta}{2}\right)}{3} (3r^2 + 3r\Delta r + \Delta r^2) \tag{11}$$





Here, $\Delta r = c\tau_{\mathrm{b}}/2$ is the range resolution of the measurement, where $\tau_{\mathrm{b}} = \tau_{\mathrm{p}}/N_{\mathrm{p}}$ is the baud length, r is the range of the volume and $\theta$ is the HPBW angle of the radar.

We assume that the noise is constant through the ion line spectrum. The noise power is then given by

$$P_{\mathrm{N}} = k_{\mathrm{B}} T_{\mathrm{sys}} B \qquad (12)$$

where $T_{\mathrm{sys}}$ is the system noise temperature, B is the bandwidth of the incoming ion-line, and $k_{\mathrm{B}}$ is the Boltzmann constant. The bandwidth we assume to be equal two times the ion thermal velocity times wave number ($v_{\mathrm{th}}k$) or the inverse of the pulse length $\tau_{\mathrm{b}}^{-1}$, depending on which one is the largest. The ion thermal velocity is given by

$$v_{th} = \sqrt{\frac{k_B T_i}{m_i}}, \qquad (13)$$

where $m_{\mathrm{i}}$ is the ion mass, which we set equal to 32 u corresponding to $O_2^+$. The system noise temperature we set to 100 K.

We can now use this to calculate the integration time of electron density measurements in the E-layer at 150 km for different range resolutions. The standard deviation requirement is set at 5%. The electron and ion temperatures we set to 400 K and 300 K, respectively. We assume a monostatic radar with frequency f = 230 MHz, HPBW $\theta$ =1°, and a transmitter with power of 5 MW. The transmitter gain for the core array we set to 43 dB and the receiver gain to 22 dB for one subarray of the imaging array. The interpulse period $\tau_{\mathrm{IPP}}$ is 2 ms and the longpulse length is 0.5 ms. The results are shown in Fig. 2.

The figure shows that the integration time decreases with increasing electron density and decreasing range resolution. This confirms the expected tradeoff between range and time resolution. If the electron density is not too low, a time resolution of a few seconds is possible. This however assumes a relatively low range resolution of 1000 - 2000 m, which still provides some useful information about the E-region plasma. When keeping a constant standard deviation, an enhanced electron density can be used to improve either the time or range resolution.

When using MIMO imaging, the core array is divided into multiple independent groups when transmitting. This provides more baselines as well as increases the maximum antenna separation. In this case, the imaging resolution will be improved by having a larger aperture. One of the challenges in this case will be separating the signals from different transmitters in the case of overspread radar targets. We assume that separate transmitters operate at the same frequency and that the transmitted signals are distinguished using radar transmit coding. This can be achieved in practice using a different pseudorandom transmit code

on each transmit group (Sulzer, 1986, Vierinen et al., in preparation). Then, the transmit power is spread over the transmitters. However, since all transmitters point in the same direction, the power adds up again. The transmit gain must be divided by the number of transmitters. It could be that because of cross-coupling between antennas, there must be buffer zones between transmitters. Then the gain decreases furthermore. On the other hand, the radar will illuminate a larger volume that contains more scatterers and so increase the received power again. In conclusion, the integration time for MIMO will be longer than for

SIMO. How long is mostly dependent on the possible cross-coupling between antennas.

The calculations do not include other echoes than from incoherent scatter or other enhancements than electron density. In the case of PMSE [e.g.,][](Urco et al., 2019) and NEIALS (e.g., Grydeland et al., 2004; Schlatter et al., 2015), the echo is





**Figure 2.** Integration time of targets in the E-region observed using the E3D core for transmit and a single 91 antenna element module for receive.

significantly stronger than for incoherent scatter. These enhancements will also make shorter integration time available, and will be more promising candidates for use of MIMO imaging.

## 3    Baseline cross-correlation

In this section we calculate the correlation between signals from two different baselines, that are transmitter-receiver-pairs. The
5    aim is to determine which baselines provide information about ionospheric features of a certain scale size, and to determine how the nearfield geometry affects this correlation.

We consider a case with one transmitter and two receivers placed with a equally long distance from the transmitter in every direction. This configuration is shown in Fig. 3. Let the transmitter be placed in the origin and the receivers at $|P_1\rangle$ and $|P_2\rangle$. Let the transmitter transmit an electrical potential of the form $V_0 = Ke^{i\omega t}$, where $K$ is a time-independent constant, $\omega$, is the



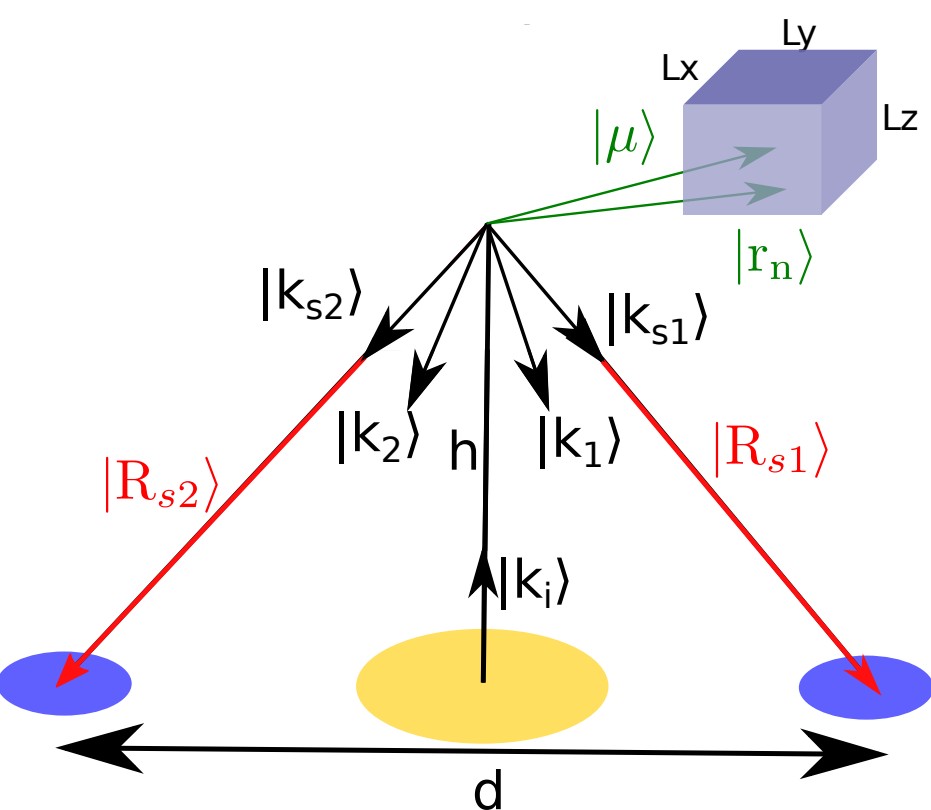

**Figure 3.** Setup for calculating the cross-correlation function. The box represents an ionospheric feature with size L. The figure is based on the assumptions in the end of Sect. 2, but is not to scale.

transmit frequency and $t$ is time. The electrical potential induced to receiver antenna $r$ then becomes

$$V_r = K \sum_{n=1}^{N} G e^{-i\omega(T_{in}+T_{srn})}, \tag{14}$$

where $T_{in} = ||R_i + r_n\rangle|/c$ is the time delay from the transmitter to scatterer $n$ and $T_{srn} = ||R_{sr} - r_n\rangle|/c$ is the time delay from scatterer $n$ to receiver $r$. Here $|R_i\rangle$ is the vector from the transmitter to the centre of the illuminated plasma volume, $|R_{sr}\rangle$ is the vector from the centre of the plasma volume to receiver $r$, and $|r_n\rangle$ is the vector from the centre of the plasma volume to scatterer $n$, like in Fig. 3. $N$ is the number of scatterers in the scattering volume, and $G \in \Re$ is the scattering gain which includes the free-space path loss. The gain may be dependent on the position of the scatterer, the scatterer itself, and on time, but we neglect these dependencies. We also neglect that the distance to the scatterer varies between the transmitter-receiver baselines. This has a order of magnitude of ~10 m, which is lower than the best available range resolution.





The cross-correlation function for time lag $\tau = 0$ can then be written as

$$R_{V_1 V_2}(t, t+0) = \mathrm{E}\left[V_1 \bar{V}_2\right] \tag{15}$$

$$= \mathrm{E}\left[K \sum_{n=1}^{N} G e^{-i\omega(T_{in} + T_{srn})} \bar{K} \sum_{n'=1}^{N} G e^{i\omega(T_{in'} + T_{srn'})}\right]. \tag{16}$$

By taking the first-order Taylor approximation of the time delays around $|r_n\rangle = |0\rangle$, we get that $T_{in} \approx \frac{R_i + \langle \hat{R}_i | r_n \rangle}{c}$ and

$T_{srn} \approx \frac{R_{sr} + \langle \hat{R}_{sr} | r_n \rangle}{c}$, where the hat denotes a unit vector. Carrying out this approximation is essentially the same as assuming plane waves. We note that $-\frac{\omega}{c}\langle \hat{R}_i - \hat{R}_{s1}| = -\langle k_i| + \langle k_{s1}| = \langle k_1|$, that is the Bragg scattering vector. Equation 16 can then be written as

$$R_{V_1 V_2}(0) = |K|^2 G^2 \sum_{n=1}^{N} \sum_{n'=1}^{N} e^{-i\frac{\omega}{c}(R_{s2} - R_{s1})} \mathrm{E}\left[e^{i\langle k_1 | r_n \rangle - i\langle k_2 | r_{n'} \rangle}\right]. \tag{17}$$

We assume that the scatterer positions are independent identical normally distributed with mean $|\mu\rangle$ and covariance $\mathbb{L}$, that is

like a Gaussian blob,

$$f_{|r_n\rangle}(|r_n\rangle) = \frac{e^{\frac{-\langle r_n - \mu | \mathbb{L}^{-1} | r_n - \mu \rangle}{2}}}{(2\pi)^{3/2}|\det(\mathbb{L})|} \tag{18}$$

We use the definition of expectation and then solve the integral. Since the positions of the scatterers are assumed to be independent, the expectation becomes zero when $n \neq n'$. In addition, in first order approximation, $R_{s2} - R_{s1} \approx 0$ because $D << h$. The result then becomes

$$R_{V_1 V_2}(0) = |K|^2 G^2 N e^{i\langle k_1 - k_2 | \mu \rangle} e^{-\frac{\langle k_1 - k_2 | \mathbb{L} | k_1 - k_2 \rangle}{2}}. \tag{19}$$

The normalized cross-correlation function,

$$\rho_{12} = \frac{R_{V_1 V_2}}{\sqrt{R_{V_1 V_1} R_{V_2 V_2}}},$$

becomes

$$\rho_{12}(0) = e^{i\langle k_1 - k_2 | \mu \rangle} e^{-\frac{\langle k_1 - k_2 | \mathbb{L} | k_1 - k_2 \rangle}{2}}. \tag{20}$$

We note that if the transmitter(s) and all receivers lay in a plane, the vertical components of the Bragg scattering vectors are exactly equal and make the vertical components of $|\mu\rangle$ and $\mathbb{L}$, namely $\mu_z$, $\mathbb{L}_{xz}$, $\mathbb{L}_{yz}$ and $\mathbb{L}_{zz}$ arbitrary. This means that the horizontal resolution is independent on the vertical resolution.

Equation 20 for $\langle \mu | = [0, 0, 0]$ and $\mathbb{L} = (L/2)^2 \mathbb{I}$ is plotted in Fig. 4, where L is the extent of the ionospheric feature in all dimensions, and $\mathbb{I}$ is the identity matrix. Figure 4 also shows numerically simulated normalized correlation based on a direct

simulation of Equation 16, which does not significantly differ from the analytical expression. The plot shows that for a height of $10^5$ m (100 km) and a baseline of 211 m, the correlation crosses 0.95 at a blob size of 70 m and 0.5 at a blob size of 250





m. At 100 km height, the radar beam of E3D is about 1800 m wide. This means that when considering a maximum baseline of 200 m and a ionospheric feature that is larger than 250x250 m, addition of longer baselines contribute less to recover the image. The E3D core has a maximum baseline of 75 m. We can simplify these calculations by setting the magnitude of the desired least cross-correlation to $\mathcal{R}$,

$$\mathcal{R} = |\rho_{12}|. \tag{21}$$

We assume that the scatterers have equal variance in x- and y-direction ($\mathbb{L}_{xx} = \mathbb{L}_{yy} = (L/2)^2$) and that all directions are uncorrelated ($\mathbb{L}_{xy} = \mathbb{L}_{xz} = \mathbb{L}_{yz} = 0$). By using the geometry as in Fig. 3,

$$\langle k_2 - k_1 | \mathbb{L} | k_2 - k_1 \rangle = \frac{4\pi^2 D^2}{\lambda^2 \left(h^2 + \frac{D^2}{4}\right)} \left(\frac{L}{2}\right)^2. \tag{22}$$

This can be compared with the cross-correlation function derived in the far-field for a zenith centered 2d Gaussian image with angular standard deviation length of $\sigma$ when measured with two antennas displaced by distance $D$:

$$
\begin{aligned}
\mathrm{E}[V_1 \overline{V}_2] &= \int\limits_{-\infty}^{\infty} \int\limits_{-\infty}^{\infty} e^{-\frac{1}{2\sigma^2}(u^2 + v^2)} e^{i\frac{2\pi}{\lambda} D u} \, du \, dv \\
&= 2\pi\sigma^2 e^{-\frac{2\pi^2 D^2 \sigma^2}{\lambda^2}}
\end{aligned}
$$

With the small angle approximation $\sigma^2 = L^2/4h^2$, and thus the term inside the normalized correlation function is:

$$\frac{\mathrm{E}[V_1 \overline{V}_2]}{2\pi\sigma^2} = e^{-\frac{4\pi^2 D^2}{\lambda^2 h^2} \left(\frac{L}{2}\right)^2}. \tag{23}$$

It should be noted that this only differs by a correction term $D^2/4$ from the near-field solution in Equation 22.

By combining Eqs. 20, 21, and 22, we get

$$\ln \mathcal{R}^2 = -\left(\frac{L}{2}\right)^2 \frac{4\pi^2 D^2}{\lambda^2 \left(h^2 + \frac{D^2}{4}\right)},$$

which can be rewritten to an expression for the feature size L:

$$L = \frac{\lambda}{\pi D} \sqrt{\left(\frac{D^2}{4} + h^2\right) \ln \frac{1}{\mathcal{R}^2}} \tag{24}$$

For short baselines or long distances $D \lesssim h/5$, the expression can be simplified, we solve for the baseline $D$ and get

$$D = \frac{\lambda h}{\pi L} \sqrt{\ln \frac{1}{\mathcal{R}^2}}. \tag{25}$$

The resulting expression shows how long the baseline can be to still get contribution to recovering the feature. Equation 25 is plotted in Fig. 5.

A longer baseline can contribute to recover smaller features, but the improvement will decrease the longer the baseline is. For example if we want to resolve a feature with size 100 m, baselines up to 200 m have large contribution to the imaging.




Adding longer baselines will improve the resolving less, and stopping slightly above 1 km . This means that the improvement of the imaging quality by including the E3D outriggers will be large for the closest outriggers. The signal received by furthest ones will correlate little with the signal received by the core. From Fig. 5 we see that the correlation in the longest E3D baseline of 1.2 km is about 5 %. This means that if one wants to use E3D to invest ionospheric features with extent around 100 m at 100

km range, there is no need to add longer baselines, the furthest outriggers are far enough. Also, it could be possible to improve the imaging quality in this example by having more baselines with lengths of around 100 m. This is one reason to use the E3D core as multiple transmitters to add more baselines.

Baselines between the receiver sites in Skibotn, Karesuvanto and Kaiseniemi are so long that they can not be used for imaging as signals will not be correlated anymore. The baseline cross-correlation calculations also do not claim that the image

is well recovered if including the largest baseline. This is more dependent on which baselines are used, how they are distributed, and how the image is recovered.

## 4   Radar imaging model

We consider a radar that may have single or multiple inputs (transmitters), and multiple outputs (receivers) (SIMO or MIMO). The radar illuminates a plasma volume at range $R$ with thickness $dr$, and inside of the one-way HPBW $\theta$. We imagine that the

volume is divided into $M$ parts, or pixels, see Fig. 6 .

The signal transmitted from transmitter A and spread by plasma element/pixel $q$ causes a voltage fluctuation in the receivers. The voltage fluctuation of receiver D due to transmitter A and plasma pixel $q$ is denoted as $V_{\mathrm{AD}}^{q} = FV_{\mathrm{A}}e^{2\pi i f T_{\mathrm{AD}}^{q}}$, where $V_{\mathrm{A}}$ is the amplitude of the signal sent by transmitter A, F is a function of the received signal amplitude, f is the radar transmitting frequency and $T_{AD}^{q}$ is the time delay of the signal due to travelling from transmitter A via pixel $q$ to receiver D, c.f. Fig. 6.

The correlation between signals from two different baselines AD and BE due to an infinitesimal scattering volume $dV$ can be described as

$$d\rho_{\mathrm{ADBE}} = \frac{P_{\mathrm{t}}G_{\mathrm{t}}(|r\rangle)G_{\mathrm{r}}(|r\rangle)\lambda^2\sigma_{\mathrm{p}}n_{\mathrm{e}}(|r\rangle)}{(4\pi)^3R_{\mathrm{t}}^2(|r\rangle)R_{\mathrm{r}}^2(|r\rangle)}e^{2\pi i f\left(T_{\mathrm{AD}}^{|r\rangle}-T_{\mathrm{BE}}^{|r\rangle}\right)}dV \tag{26}$$

where $P_{\mathrm{t}}$ is transmit power, $G_{\mathrm{t}}$ is transmit gain, $G_{\mathrm{r}}$ is receiver gain, $\lambda$ is the radar wavelength, $\sigma_{\mathrm{p}}$ is the scattering cross-section for a single electron given by Eq. (9), $n_{\mathrm{e}}$ is the electron density, $R_{\mathrm{t}}$ is the distance from transmitter to the scattering volume, $R_{\mathrm{r}}$

is the distance from the scattering volume to the receiver, and $|r\rangle$ is the position of the scattering volume. We integrate over the whole scattering volume to get the whole measurement. At a certain time lag, we get the correlation for the range of interest. We assume that the gains are constant inside of the radar beam and zero otherwise and neglect the dependency of $R_{\mathrm{r}}$ and $R_{\mathrm{t}}$ on the exact position of the scattering volume. The correlation can then be written as

$$\rho_{ADBE} = \frac{P_tG_tG_r\lambda^2\sigma_p}{(4\pi)^3R_t^2R_r^2}\int_V n_e(|r\rangle)e^{2\pi i f\left(T_{\mathrm{AD}}^{|r\rangle}-T_{\mathrm{HB}}^{|r\rangle}\right)}dV. \tag{27}$$

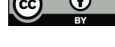



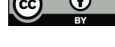



**Figure 4.** Cross-correlation between signal in EISCAT 3D receivers displaced by distance $d$. The solid lines show the magnitude of the normalized cross-correlation function, Eq. (20) with $\mathbb{L} = (L/2)^2\mathbb{I}$. The dots show numerical estimations of the cross-correlation, Eq. 16.



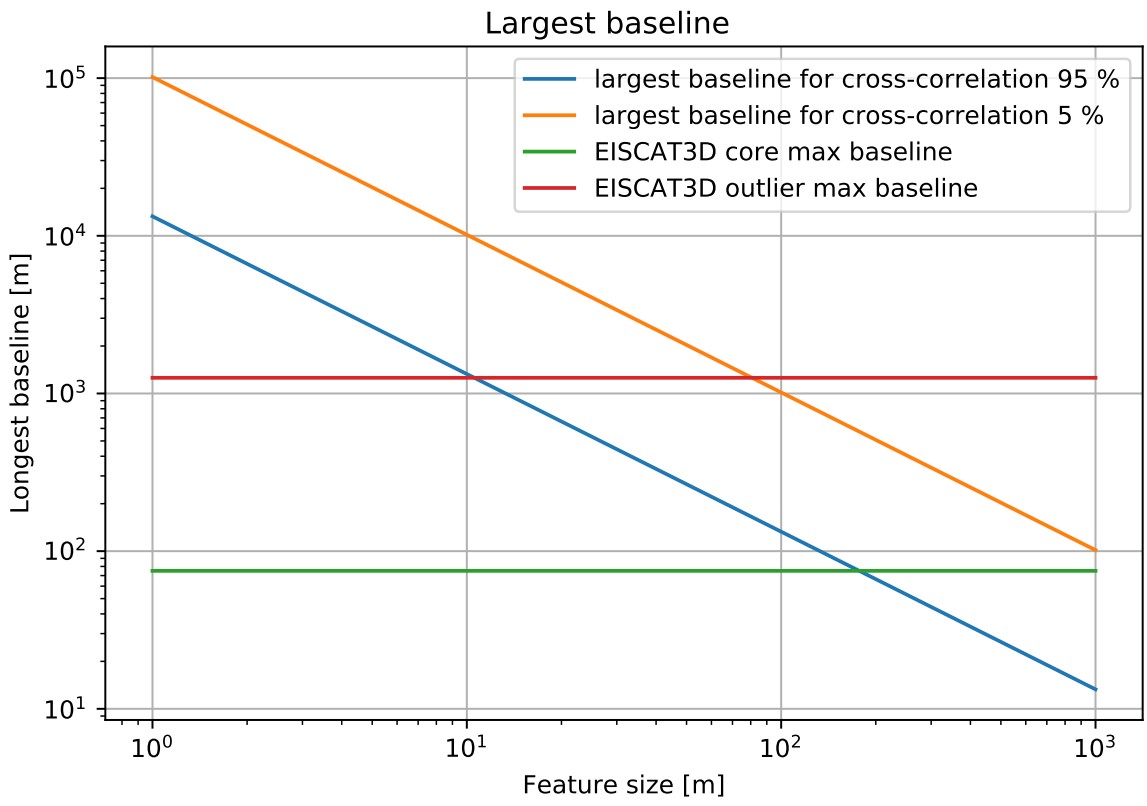

**Figure 5.** Largest baseline for recovering ionospheric features of certain size. Measurements in the area under the blue line have high (>95 %) correlation, over the orange line the correlation is lower than 5 % . Longer baselines cannot be used to resolve features of this size.

We assume that the electron density (or brightness) distribution can be written as a sum of discretized parts with constant electron density. We neglect variations in the phase shift inside of one part. The integral can then be replaced with a sum

$$\rho_{\mathrm{ADBE}} = \frac{P_{\mathrm{t}} G_{\mathrm{t}} G_{\mathrm{r}} \lambda^2 \sigma_{\mathrm{P}}}{(4\pi)^3 R_{\mathrm{t}}^2 R_{\mathrm{r}}^2} \mathrm{d}r \left(2R\tan\frac{\theta}{2}\right)^2 \sum_{q=1}^{Q} \frac{n_{\mathrm{e}}[q]}{Q} e^{2\pi i f\left(T_{\mathrm{AD}}^q - T_{\mathrm{HB}}^q\right)}. \tag{28}$$

The first factor here is constant and can be normalized away. The number of discretizations $Q$ is still needed in the simulations
5 if the original image has an other resolution than the reconstructions. The series of measurements can be written on matrix form,

$$|m\rangle = \mathbb{A}|x\rangle + |\varepsilon\rangle. \tag{29}$$



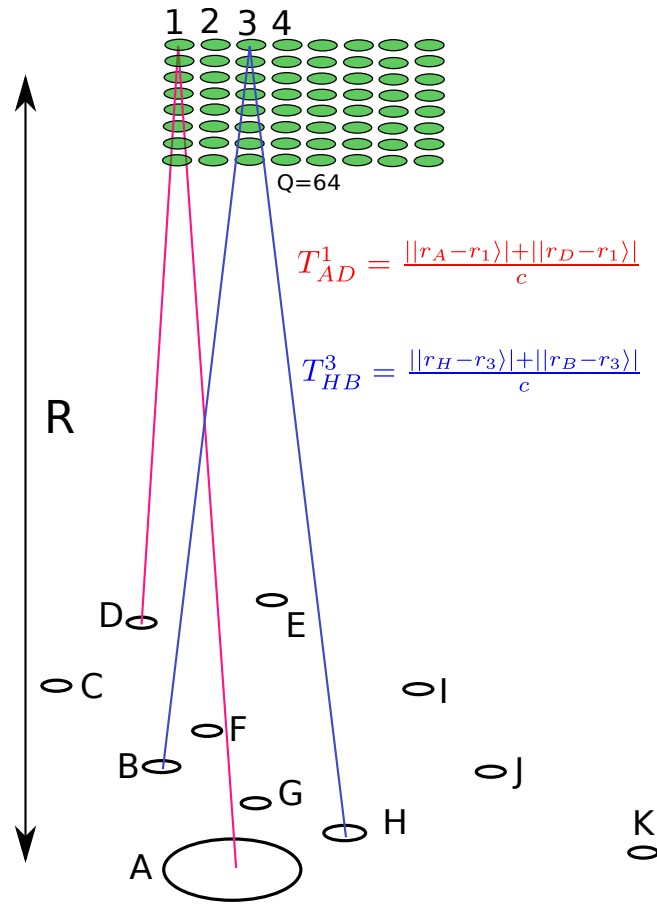

**Figure 6.** Example of multiple input multiple output radar and plasma volume in its line of sight.

Here, $|x\rangle = [n_e[1], n_e[2], \ldots, n_e[Q]]^T$,

$$\mathbb{A} = \begin{bmatrix} e^{2\pi i f\left(T_{AA}^1 - T_{AA}^1\right)} & \cdots & e^{2\pi i f\left(T_{AA}^Q - T_{AA}^Q\right)} \\ \vdots & \ddots & \vdots \\ e^{2\pi i f\left(T_{KK}^1 - T_{KK}^1\right)} & \cdots & e^{2\pi i f\left(T_{KK}^Q - T_{KK}^Q\right)} \end{bmatrix}$$

is the theory matrix, and $|\varepsilon\rangle = [\varepsilon_{AAAA}, \varepsilon_{AAAB}, \ldots, \varepsilon_{KKKK}]^H$ is the complex normally distributed noise vector.





Sometimes it is more convenient to have the crosscorrelations on matrix form. The measurements can be transferred from the one form to the other simply through reshaping the vector $|m\rangle$ to a matrix $\mathbb{M}$ or opposite:

$$\mathbb{M} = \begin{bmatrix} \rho_{\text{AAAA}} & \rho_{\text{AAAB}} & \cdots & \rho_{\text{AAKK}} \\ \rho_{\text{ABAA}} & \rho_{\text{ABAB}} & \cdots & \rho_{\text{ABKK}} \\ \vdots & \vdots & \ddots & \vdots \\ \rho_{\text{KKAA}} & \rho_{\text{KKAB}} & \cdots & \rho_{\text{KKKK}} \end{bmatrix} \tag{30}$$

To get an estimate of the intensities of the plasma in the image, Eq. (29) has to be inverted so that

$$|\hat{x}\rangle = \mathbb{B}|m\rangle, \tag{31}$$

where $\mathbb{B}$ is a matrix that reconstructs the image $|x\rangle$ from the measurements $|m\rangle$. When inserting Eq. (29) into Eq. (31) and neglecting noise, we get $|\hat{x}\rangle = \mathbb{B}\mathbb{A}|x\rangle$. We wish that the reconstructed image is as close to the reality as possible, and so taking $\mathbb{B} = \mathbb{A}^{-1}$ would give a perfect solution. However, in the most cases will $\mathbb{A}$ not be invertible, mostly because it is not a square matrix, and the exact inverse will therefore not exist. Other attempts are therefore needed.

## 4.1 Matched filter

When the scatterers are behind the Fraunhofer limit in the far field, Eq. (29) represents a Fourier transform. One approach to get back the original image would be the inverse Fourier transform, which can be represented the hermitian conjugate of the theory matrix, $\mathbb{B} = \mathbb{A}^{H}$ like a matched filter (MF). Unfortunately, the samples of the Fourier-transformed image, that are the visibilities, are sparsely and incomplete scattered and the problem gets underdetermined (Hysell and Chau, 2012; Harding and Milla, 2013). The approach can be interpreted as steering the beam after the statistical averaging and is therefore also called beamforming.

## 4.2 Capon method

Another approach is the Capon method (Palmer et al., 1998). The purpose of this method is to minimize the intensities in all other directions than the direction of interest, that is to minimize the sidelobes of the antenna array in directions with interfering sources. The result is to invert the matrix of correlation measurements $\mathbb{M}$ (Palmer et al., 1998). In order to continue using the notation in this article, $\mathbb{M}^{-1}$ is reshaped back to a vector $|m^{-1}\rangle$. The estimated intensities by the Capon method can then be written as

$$|\hat{x}_{capon}\rangle = \frac{1}{\mathbb{A}^{H}|m^{-1}\rangle}, \tag{32}$$

where the fraction denotes element-wise division.

## 4.3 Singular value decomposition

The problem in Eq. (29) is overdetermined if the number of unknowns, that is the number of discretizations, is less than the number of measurements. This can be the case if we solve for a imaging resolution that is low enough. We can then use the





method of least squares to solve it, getting

$$|\hat{x}_{LS}\rangle = \left(\mathbb{A}^{\mathrm{H}}\mathbb{A}\right)^{-1}\mathbb{A}^{\mathrm{H}}|m\rangle.$$

One can also use the singular value decomposition (SVD) on the theory matrix, $\mathbb{A} = \mathbb{U}\mathbb{S}\mathbb{V}^{\mathrm{H}}$, where $\mathbb{S}$ is a diagonal matrix containing the singular values, that are square roots of the eigenvalues of $\mathbb{A}^{\mathrm{H}}\mathbb{A}$, $\mathbb{V}$ contains the normalized eigenvectors of

$\mathbb{A}^{\mathrm{H}}\mathbb{A}$, and $\mathbb{U}$ contains the normalized eigenvectors of $\mathbb{A}\mathbb{A}^{\mathrm{H}}$. The inversion matrix $\mathbb{B}$ can then be written as

$$\mathbb{B} = \mathbb{V}\mathbb{S}^{-1}\mathbb{U}^{\mathrm{H}}, \tag{33}$$

which can be shown still gives the same solution as ordinary least squares, but with increased numerical accuracy (Aster et al., 2013). Because of inverting the singular values, the eigenvectors corresponding to the smallest values contribute most to the variance of the solution and make the solution sensitive to noise. Also, the problem can be rank deficient, that is that several

columns in the theory matrix are nearly linear dependent on each other. The problem is then said to be ill-conditioned or multicollinear.

In such cases, some singular values will be practically zero and the solution may be hidden in the noise. To prevent the noise sensitivity, the solutions can be regularized. This makes the reconstruction biased towards smoothness and zero, but less noisy (Aster et al., 2013). We here consider two regularization techniques, truncated SVD (TSVD) and Tikhonov regularization. In

TSVD, the inverse of the singular values below some limit are set to zero. The eigenvectors corresponding to the smallest singular values will then not contribute to the result. These eigenvectors often contain high frequency components. Ignoring them makes the solution smoother. Tikhonov regularization or ridge regression can be done in several ways. In this article, we use zeroth order Tikhonov regularization where the singular values $s_i$ are inverted with

$$\frac{s_i}{s_i^2 + \alpha^2}, \tag{34}$$

where $\alpha$ is a regularization parameter. By using SVD, we also can get the variance $|\Sigma_{\hat{x}}\rangle$ of the estimates. For pure least squares, it is $\mathrm{diag}((\mathbb{A}^H\mathbb{A})^{-1})$, for regularized least squares it is

$$|\Sigma_{\hat{x}}\rangle = \mathrm{diag}(\mathbb{B}\mathbb{B}^H). \tag{35}$$

### 4.4 CLEAN

CLEAN is another attempt to reduce sidelobes. It is based on the matched filter approach, but iteratively finds the real structure

in the field of view (Högbom, 1974). It supposes a source where the image reconstructed by the matched filter is brightest. The source is added to an image that only is containing the suspected sources, which will be the reconstructed image. Then the measurements that the radar would have measured if the reconstructed image were the true image are subtracted from the real measurements and the next suspected source is found. This procedure is repeated until there are no clear sources left in the measurements (Högbom, 1974). The method is a special case of compressed sensing requires an assumption on how the

measured sources look like (Harding and Milla, 2013). For sparse sources, a Dirac delta function could be appropriate, but lead to sparse solutions.





## 4.5 Performance of the radar layouts

We considered different radar layouts. The layouts together with plots of the visibilities and the point spread function are shown in Fig. 7.

When considering a layout with multiple transmitters and multiple receivers (MIMO), it is assumed that the signals from different transmitters can be distinguished. This increases the number of virtual receivers and thereby the visibilities get more widespread and denser, cf. Fig 7. However, using multiple transmitters increases the integration time as described in Sect. 2. We note that when receiving with the outriggers, the main beam becomes narrower. Also, there are gaps in the visibilities. This is due to the sparse locations of the outriggers and makes the point spread function look more irregularly (cmp. Fig 7c and 7f). With multiple transmitters, the main beam becomes even narrower (cmp. Fig 7c and 7i). When both using multiple transmitters

and receiving outriggers, the gaps in the visibility domain partially get filled and the sidelobes are clearly reduced. The MIMO layout used here could possibly be improved by using positions of the transmitters so that gaps in the visibility get more filled.

## 4.6 Performance of the imaging techniques

We simulate E3D measurements using Eq. 29 and with the presented antenna configurations. As original image, we use a part of Fig. 1. From the measurements we reconstruct the images with the matched filter (MF), Capon, truncated singular value

decomposition (TSVD) and CLEAN techniques. For TSVD, the singular values below 0.02 of the maximum singular value were truncated. This value gave the best combination of resolution and low noise level (regularization). For CLEAN, we used a gain of 1 and a threshold of 1.36 times the average value. We tried both Dirac delta and Gaussian functions in the CLEAN kernel. In capon filtering, it happens that the correlation measurement matrix $\mathbb{M}$ is singular. In such cases, TSVD is used to invert the matrix. This truncation ignores singular values that are less than 0.03 % of the largest singular value.

To all cross-correlations there is added white complex Gaussian noise with zero mean and 5 % standard deviation. The noise is equal for every reconstruction of a single resolution, but varies between reconstruction in different resolutions. The results for the SIMO layout is shown in Fig. 8.

    Of the reconstruction techniques, TSVD clearly gives the best results. It is also the only method that fairly reproduces the shape of the true image. Capon also partly reproduces the shape, but far worse than TSVD. The matched filter apparently only

reproduces something similar to the point spread function. The performance of CLEAN (not shown here) is accordingly poor. In terms of calculation time, CLEAN is the slowest algorithm followed by TSVD. MF and Capon are relatively fast. These differences get stronger when also considering MIMO. Most of the computation time of TSVD is used to invert the theory matrix. Since the theory matrix only varies from experiment to experiment, it must only be inverted once and can be saved afterwards. The computation time therefore is reduced to a simple matrix multiplication, and it is not considered as a problem

for the real radar. We therefore concentrate on images reconstructed with techniques using SVD. Here, we compared ordinary least squares, TSVD with truncating singular values under 2 %, like before, and Tikhonov regularization with regularizing parameter $\alpha = 10$ and 100. These results are shown in Figs. 9 to 12 for the four layouts shown in Fig. 7, that are SIMO without and with outriggers and one MIMO case without and with outriggers, respectively.



**Figure 7.** E3D transmitter-receiver layouts considered. The left column shows the layouts, the middle column shows the visibilities, and the right column shows the point spread function in the near-field at 100 km range. The point spread function was calculated by reconstructing a 1x1 one-valued central pixel in an 129x129 zero-valued pixel image with a matched filter. The upper layout uses the whole core array as a single transmitter and receives with each of the 109 antenna groups in the core array. The second layout also includes the interferometric outriggers. In the third layout only the core is used, but divided into three transmitters. Finally a fourth layout uses both the outriggers and multiple transmitters

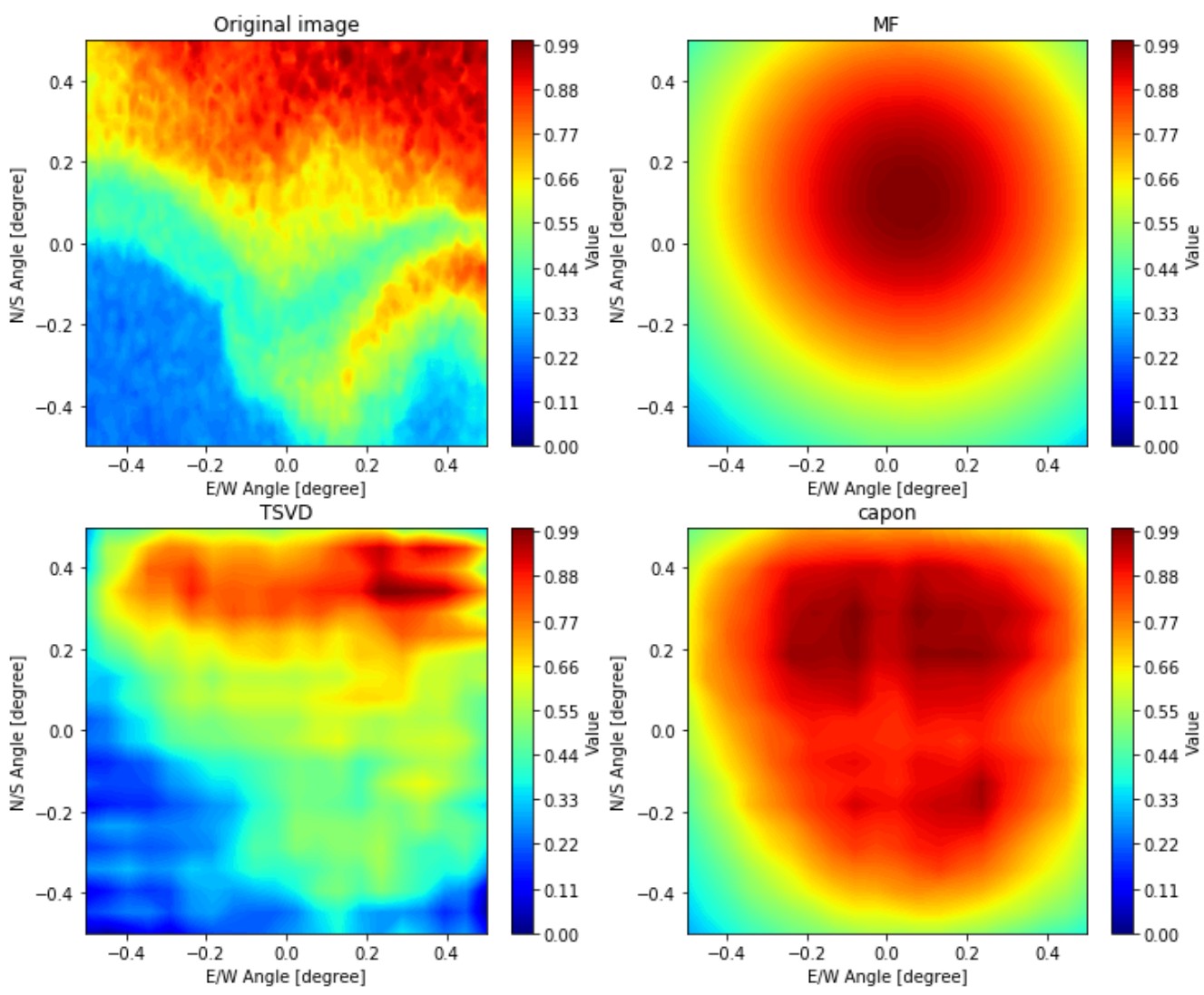

**Figure 8.** Comparison of reconstructions. All antennas are transmitting together like one transmitter, but receiving separately. The intensities are normalized to be between 0 and 1. The top-left figure shows the true image. The others show the reconstructed image: Matched filter (top-right), TSVD (bottom-left), Capon (bottom-right). For TSVD, the singular values below 0.02 of the maximum singular value were ignored.



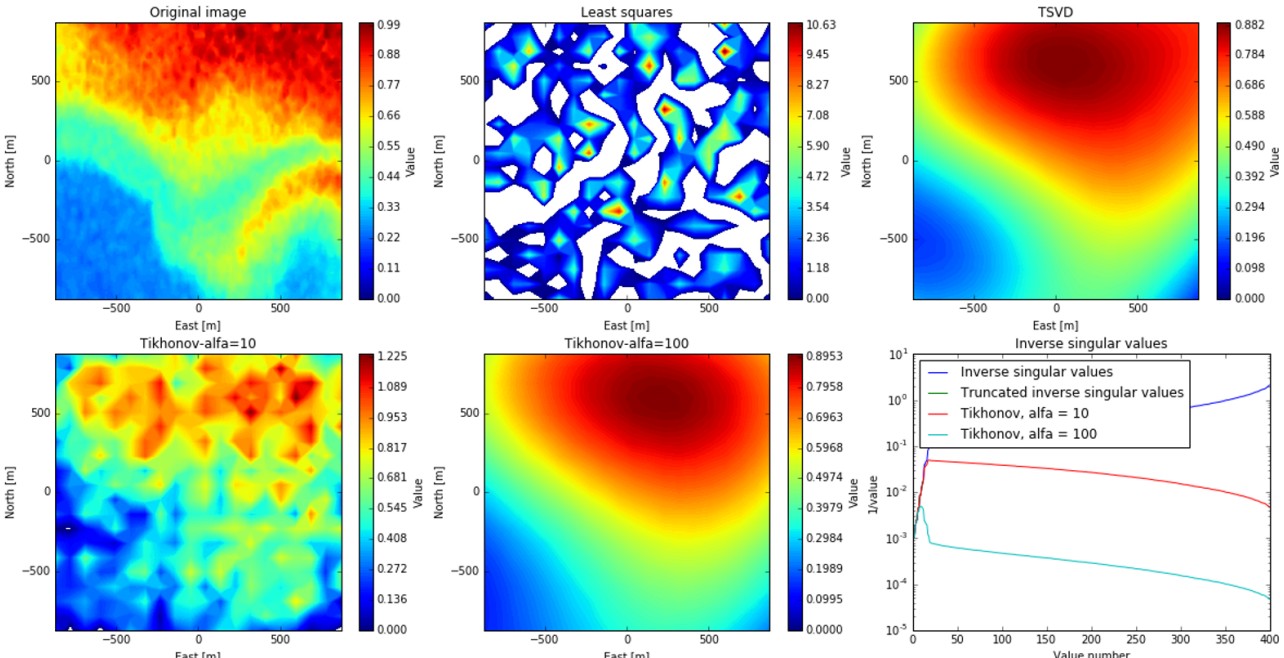

**Figure 9.** Comparison of reconstructions using SVD for the SIMO case only using the core antennas. The topleft figure shows the true image. The bottomright image shows the inverse of the singular values of the theory matrix. The other figures show reconstructions with different weighting of the singular values: The topmiddle has no weighting and corresponds to ordinary least squares method, in the topright figure, the singular values below 2 % of the largest singular value are ignored/truncated away, in the bottomleft and bottommiddle, the inverse of the low singular values are damped like in Eq. (34) with regularizing parameter $\alpha$ of respectively 10 and 100. The reconstructions consider a image resolution of 20x20 pixels, at 100 km altitude 1 pixel corresponds to 100x100 m. White spaces in the color plots correspond to negative values.

For all layouts, at a considered resolution of 20x20 pixels in the radar main beam (here, 1 pixel≈100x100 m), the image reconstruction with method of least squares is very noisy. The singular values of $\mathbb{A}$ are varying over several orders of magnitude which is a sign of that columns in $\mathbb{A}$ are linear dependent on each other. The regularized solutions look considerably better, with a regularization parameter of 10, the recovered images are still a bit noisy, but with stronger regularization the images get

5    smoother and closer to the original.

The two most strongly regularized images to the upper-right and bottom-middle contain stripes if the radar layout is including the outriggers. This is propably because the visibility in some regions has gaps, cf. Fig 7. When only considering the core array, there are no gaps other than the spacing between antennas. The recovered images without the outriggers look smoother than including the outriggers, but when including the outriggers, more details of the original image can be seen. Also, in the MIMO

10    case with outriggers, the feature in the south-east can be seen in the reconstruction. For the other layouts it is less visible and not clearly distinguishable from the main feature in the north.

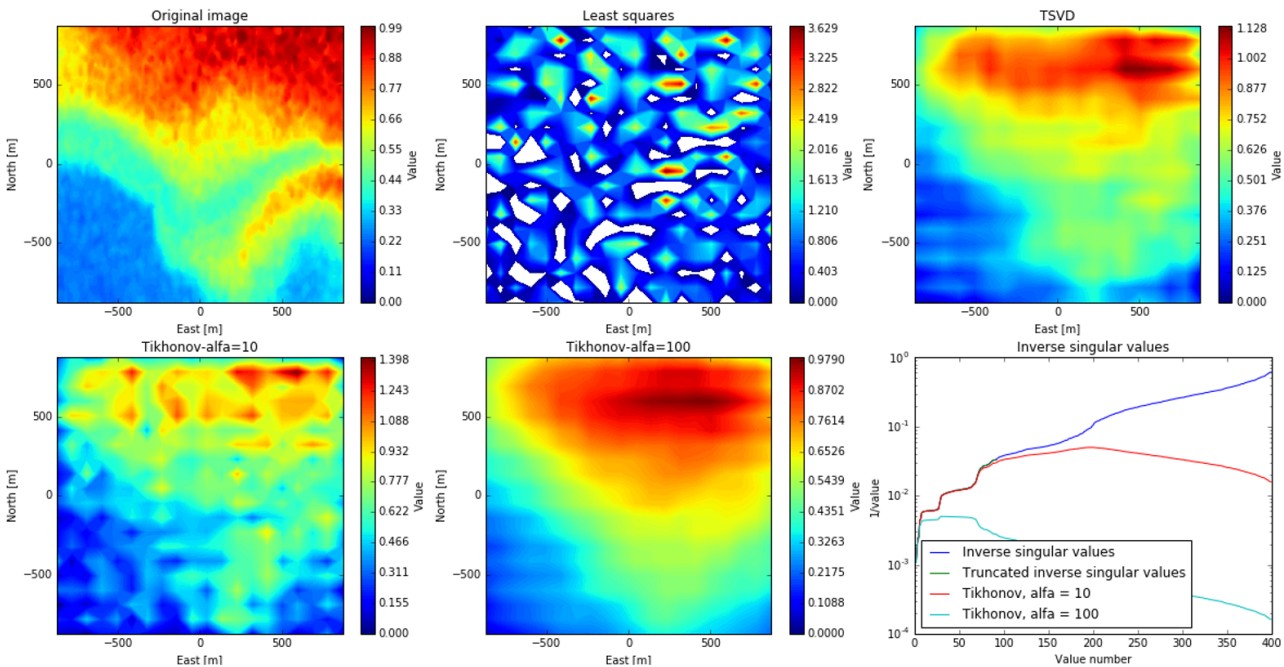

**Figure 10.** Comparison of reconstructions using SVD for the SIMO case including the outriggers. Else, the plots corresponds to Fig. 9.

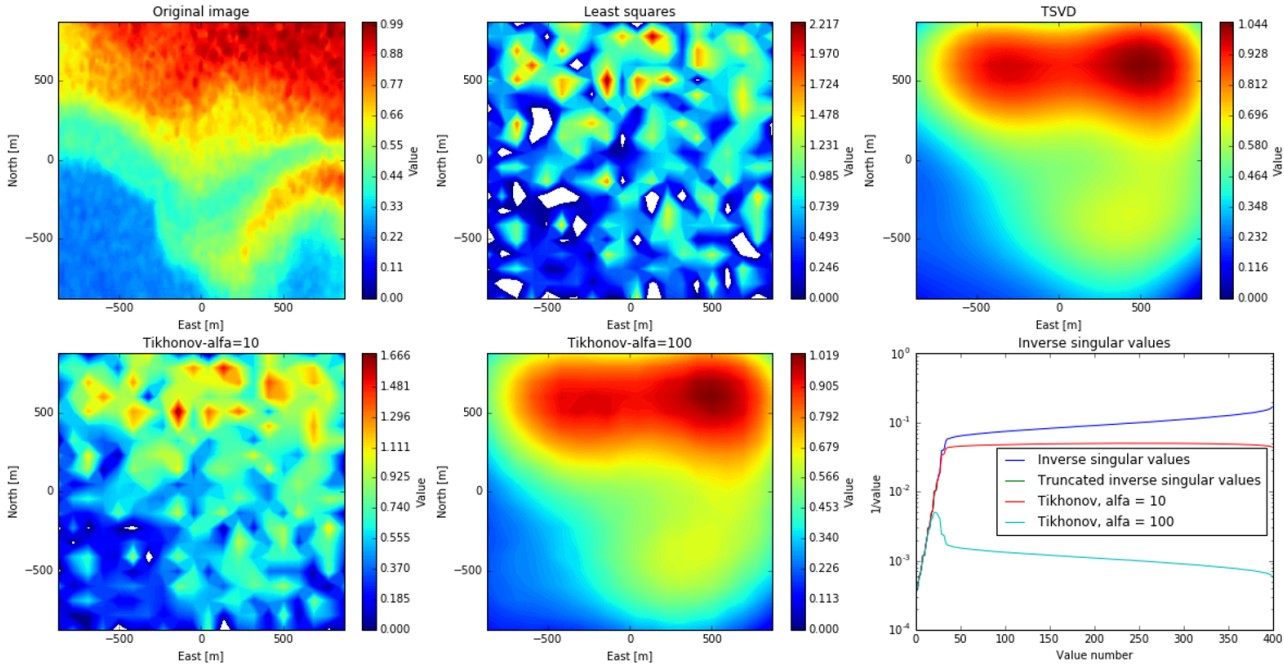

**Figure 11.** Comparison of reconstructions using SVD for the MIMO case only using the core antennas. Else, the plots corresponds to Fig. 9.

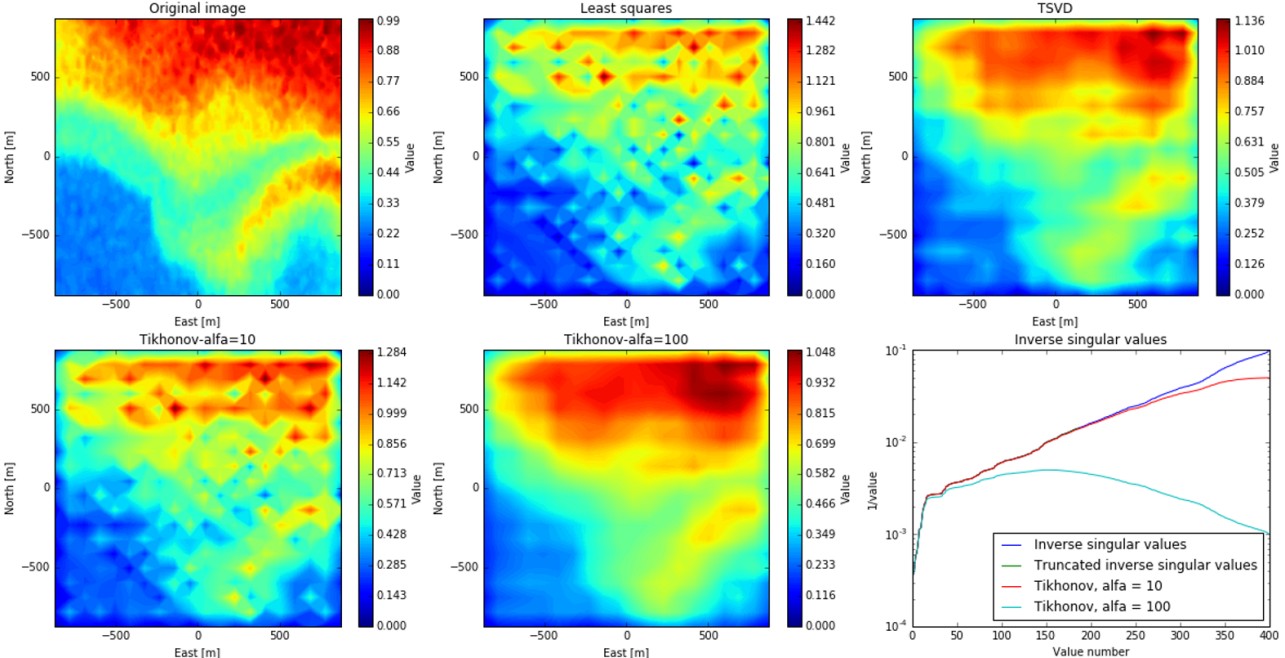

**Figure 12.** Comparison of reconstructions using SVD for the MIMO case including the outriggers. Else, the plots corresponds to Fig. 9.

The uncertainty of the reconstruction itself is given by the variance of the recovered image, Eq (35). The mean standard deviation for the different layouts and reconstruction techniques is shown in Fig. 13. The plots of the least square variance are comparable to the variance plots in Lehtinen (2014). We note that while Lehtinen (2014) investigates far-field imaging, Fig. 13 shows near-field imaging.

5    By using the standard deviation we neglect errors introduced by the discretization because they are not included in the variance. This assumption is true if the true image has the same resolution as the reconstruction, but that is only for the case of what Kaipio and Somersalo (2010) call a "inverse crime". In reality, the target of E3D, the electron fluctuations in the ionosphere, is not discrete with steps of several metres. Also, by regularization, bias is introduced to the solution, which the variance does not take into account. Therefore, we also used the similarity to the true image for uncertainty estimation. As

10    a measure, we used the mean square deviation, $s = \sum_{i=1}^{N} \frac{(\hat{x}_i - x_i)^2}{N}$ so a low value of $s$ means great similarity. Because the original image and the reconstruction have different resolutions, the smallest is scaled up. The scaling was done by Lanzcos resampling with a $\cos^2$-kernel. A drawback with the MSE is that it could be influenced by the target, while the variance is not. The mean standard deviation and the similarity to the original image are shown in Fig. 13 for all layouts considered here and reconstruction resolutions up to 100x100 pixels.

15    The variance of the recovered image is strongly increasing with the resolution we assume/want that the radar target has. Image recovering with LS gives the highest variance for all layouts. The reason is the multicollinearity in the theory matrix which amplifies the noise in the recovered image. At small resolution, the variances are equal for the different reconstruction

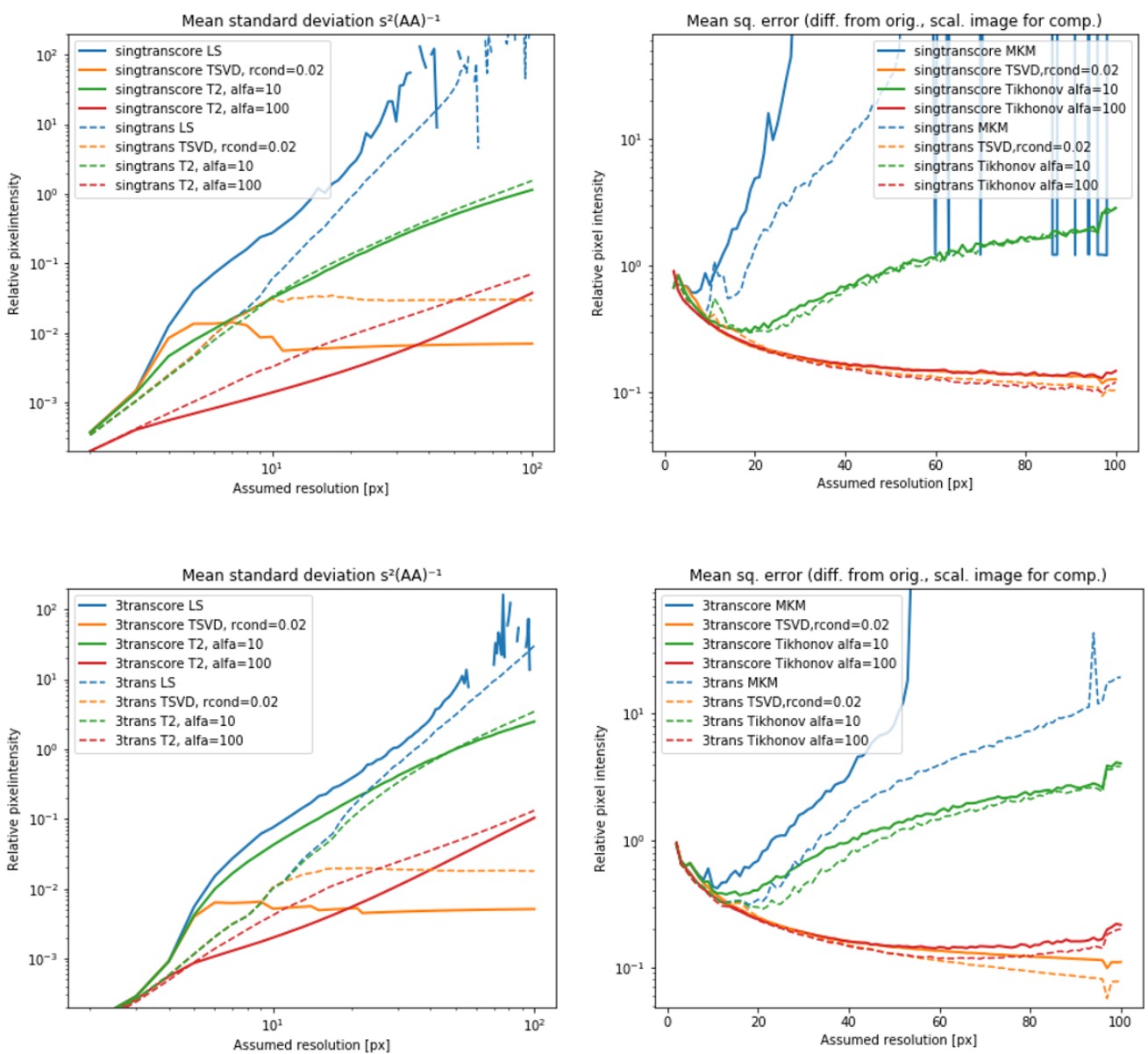

**Figure 13.** Comparison of regularization techniques. Mean standard deviation of the result is shown to the left. The right plots show the similarity of the recovered image to the true image. Both are shown relative to the mean intensity of the original image. The upper plots show the relative standard deviation and the similarity for the SIMO cases and the lower plots for the MIMO cases. The solid lines show recovers when only using the core array, the results with dashed lines include the outriggers. The line color shows the type of regularization; blue is not regularized (ordinary least squares) orange is TSVD including only singular values higher than 2 % of the greatest singular value, green and red lines are Tikhonov damped singular values with $\alpha = 10$ and 100, respectively.





techniques, but diverge when the regularization starts to influence the results. This divergence happens later when including the outrigger antennas and also later when using MIMO than for SIMO. For high resolutions, the variance of the TSVD solution is the lowest. However, since bias is introduced by regularizing the solution, this does not necessarily mean that TSVD solution is the best.

The mean square error (MSE) of the recovered images is in general higher than their mean variance. For small resolutions, it decreases with increasing resolution until it reaches a bottom point. The error then increases again. For LS and Tikhonov with $\alpha = 10$, the minimum is at 10-20 pixels per direction. When including the outriggers, the minimum is at a later stage. Also, the error is lower. We also note the dip of error at 97x97 pixels. This is exactly where the resolution of the recovered image matches the resolution of the original image so these dips are the effect of inverse crimes and therefore not transferable to the

real radar. For high resolutions, the MSE is higher for MIMO than for SIMO when using Tikhonov. This could indicate that for MIMO, more regularization is required.

  The original image contains values between 0 and 1 m$^{-3}$ with a mean of about 0.5 m$^{-3}$. In real, the values will be far higher and the uncertainty will increase accordingly. Therefore, the standard deviation and the MSE are plotted relative to the mean value of the original image. In order to have a good recover, the relative mean error should be below 1 and, if possible, far

below that. All regularized solutions would fulfil this criterion, but the two strongest regularizations clearly have the lowest MSE. The minimum of MSE seems to be somewhere between 60x60 pixels for MIMO and 90x90 pixels for SIMO. In practice, the image reconstructions for higher resolutions look very similar to low resolution (20x20) without adding more details but with better quality of the reconstructed image.

  In the MSE plots, the curves flatten out to a minimum relative MSE at about 10 %. At 100 km range, 20x20 pixels corre-

sponds to a resolution of around 90x90 m. The TSVD indicates that the recovered image with MIMO could be improved with stronger Tikhonov regularization, but this has not been investigated.

  The MSE of TSVD does not decrease significantly from SIMO to MIMO. Therefore it seems that there is little gain by using the MIMO layouts considered here. However, the feature in the bottom right part of the image in Figs. 10 and 12 gets clearer with MIMO. For other targets, these results may look different. When comparing the point spread functions in Fig. 7, it could

be that MIMO configuration is better for point-like targets, like space debris or meteors, but this is beyond the scope of this article.

  In this article, the MIMO approach to ISR and E3D has only been treated superficially. There are still some questions that must be answered. To distinguish between the transmitters, we assumed code diversity. However, there is need to study how well the signals can be distinguished. This can influence the possible number of transmitters. The placing of the transmitters

was not investigated here, the example in this article is a simple proposal. At the same time, the transmitter locations can have great influence on the visibility coverage. Also, the SNR and integration time calculations for MIMO would need to be investigated more thoroughly.





# 5 Conclusion

In this article, we have studied the temporal and spatial resolution of the upcoming E3D radar in the case of aperture synthesis radar imaging, primarily focusing on the feasibility of imaging the incoherent scatter radar return from the E-region. The most up to date radar design specifications at the time of writing this article was used as a basis of this study.

We find that the range and time resolutions are dependent on each other. When keeping the uncertainty level constant, a better range resolution goes on the cost of the time resolution. With an increase in the electron density, the resolution in time and/or range can be improved without increasing the noise level. Under normal conditions in the E-layer ($T_e \approx 400$ K, $T_i \approx 300$ K, $n_e \approx 10^{11}$ m$^{-3}$), with a desired integration time of 10 seconds, the achievable range resolution is slightly more above 1500 m.

The horizontal (imaging) resolution depends on the radar layout and the imaging technique. The imaging techniques that were evaluated were: matched filter, least squares using singular value decomposition without and with regularization, Capon, and CLEAN. Of these techniques, only regularized least squares gave satisfactory results. The two regularization techniques of either truncating or damping of the inverse singular values both worked and gave similar results.

These image reconstructions can be reduced to a simple matrix multiplication by saving the inverted theory matrix. Regularized SVD is therefore is among the fastest reconstruction techniques amongst the ones evaluated. With Tikhonov regularization with a damping coefficient of 100, or truncating away singular values below 2 % of the largest value, the relative error of the recovered image can go down to 10 %. The resolution of the recovered image is about 60x60 pixels, at 100 km range this corresponds to 30x30 m, but features smaller than 90x90 m will be blurred out.

The simulation results show that using the outriggers increases the imaging accuracy. Dividing the core array into multiple transmitters to get a MIMO system seems to increase the imaging resolution if the target is smooth. MIMO also has the drawback that it needs stronger signals or more integration time to keep the same measurement accuracy as SIMO. However, this needs further investigation. as MIMO may be useful for very bright targets such as PMSE, as well as point-like targets like space debris or meteors, but the latter needs further investigation.

We conclude with that radar imaging with EISCAT 3D is feasible.

*Acknowledgements.* J. Stamm and J. Vierinen would like to thank the Tromsø Science Foundation for supporting this work. J.S. thanks Harri Hellgren for providing information about EISCAT3D design. The EISCAT association is founded by research organizations in Norway, Sweden, Finland, Japan, China and the United Kingdom. J.S. carried out the radar simulations and calculations and prepared the manuscript. J.V. suggested the topic and supervised the project. M.U., B.G., and J.C. gave advises on decorrelation and on imaging. All authors discussed the work and participated in preparing the manuscript. J. Vierinen is editor of the editorial board of the journal for this special issue.



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
