# Peer review of "Radar Imaging with EISCAT 3D"

_Annales Geophysicae, 2020_

## Referee Comment (RC1) · Anonymous Referee #1 · 6 Jul 2020

The paper has some interesting ideas and potential for the incoherent scatter radar (ISR) community. This paper is jumping to apply MIMO radar techniques to ISR and makes a good case that this can be applied to the E-region of the ionosphere. It argues that Eiscatt 3-D will be able to do some interesting high resolution imaging of the E-region ionosphere. The authors simulate the MIMO radar systems and then find that the SVD seemed to give the best results.

The application of generalized inverse techniques to geospace sensors is something I generally support. I think this paper does a good job at taking specific problem in ISR and tries to apply general inverse theory and gets some promising results. Overall the authors did a good job of arguing that this sort of techniques will be useful for ISR. I think they need add some caveats to clarify the applicability of the technique.

First off the the technique the authors show does not include fitting for ISR spec-

tra/ACFs to get plasma parameters. This is fine if they specify that this for the E-region only and that is not clear from the title or the abstract. Yes, different techniques are required for different regions of the ionosphere with ISR but it would help the reader understand the current limits of the method you are applying!

I want to also point out there has been work on imaging ISR in the past such as seen in [1], but this was a study that only applied interpolation techniques to ISR data. I think comparing this 2010 paper with the methods here would be helpful and show that this work is new.

The following is a list a technical corrections I've found will need to fixed:

- Page 6 line 25: Do mean truncated cone? Conic section implies a 2-D plane and this is a 3-D volume. - Page 7 line 1: You state range, which range is this because we're not in a multistatic set up and the term range is ambiguous, transmit, receive, bistatic? - Page 7 lines 9-10: $O_2+$ is used for the calculation but the dominate species in the E-region is $NO+$. I don't think this will create a huge error in the calculations but it should be mentioned that this will not throw off the calculation by much. - Page 8 Figure 2: I'm having trouble seeing the need for this figure since the assumption is for a monostatic system, and chemical composition is off too. I'm aware with MIMO that the resolution can be chosen but this link budget only accounts for a monostatic system. Plus in a monostatic system the gain from the antenna is directly tied to the resolution. - Page 13 lines 13-14: Please be more clear about what was done to the image from Figure 1. It seems you just took the magnitude of the image and mapped it from 0 to 1?

Overall, good paper. I think it should be accepted with the changes I've go over above.

[1] J. Semeter, T. Butler, C. Heinselman, M. Nicolls, J. Kelly, and D. Hampton, "Volumetric imaging of the auroral ionosphere: Initial results from PFISR," J. Atmos. Solar-Terrestrial Phys., vol. 71, no. 6–7, pp. 738–743, May 2009.

[Figure]

2020.

---

## Referee Comment (RC2) · Anonymous Referee #2 · 2 Aug 2020

The present manuscript analyzes the problem of radar imaging in 3D for incoherent scatter applications that will be implemented using the EISCAT 3D radar. It is mentioned that the proposed technique includes "near field" effects on the formulation of the radar imaging problem because EISCAT 3D applications will be in such regime. The analysis includes also the concept of MIMO radars in order to improve the resolution of radar images. The manuscript is well organized and the results are presented clearly. Although the analysis performed in this document introduces new ideas related to the radar imaging problem, I would recommend a careful revision of the document before its possible publication. As I will explain there are some important issues that have to be addressed first.

1. In Equation 2 (page 5), it is assumed that the number of independent measurements per second is proportional to the number of lagged products in a longpulse experiment. This is definitely not the case. In a long pulse experiment, lag products are not independent, all of them are correlated. This is because, within the length of a pulse, signals from a common volume are mixed. Assuming that all lag products are equally informative, it is also an oversimplification that I think may lead to not necessarily correct conclusions, particularly, in this application in which the target fills the radar volumen. I would recommend the authors to review this section in order to analyze more carefully the relationship between the number of samples within a longpulse and the integration time needed to reduce statistical uncertainty. Notice that if you consider Np=1, there is a singularity in equation (7), I don't think this is correct. I would also recommend to review equation 11 since a radar volume can be modeled better as a spherical cone section rather than as a truncated cone. In this expression, if you consider "r" at the center of the radar volume, the expression becomes simplified.

2. In the introduction (line 22, page 4), it is mentioned that there is not much literature related to 3D imaging and the authors make reference to a recent work of one of the coauthors. This is not fully true, the works of Palmer et al(1998), Yu et al (2000), and Chau & Woodman (2001) (see references below) addressed the imaging problem in 3D in the same sense as the present manuscript does. Of course the difference is that the new approach is addressing the incoherent scatter problem while the previous work was mainly focussed on coherent scatter echoes. So proper references should be used.

3. In line 29, page 7, the integration time for MIMO applications is analyzed and it is mentioned that the integration time will be longer in the MIMO case than in the SIMO case, but the authors indicate that the difference depends on cross-coupling between antennas. I don't think this conclusion is correct, at least not as a first approximation. There is plenty of literature related to soft-target radar equations that explain clearly that the received power is directly proportional to an effective antenna aperture area (which is also proportional to the true antena area). So, even if you use the same power on transmission, the received power will be less when using a small antenna. Then, the need for additional integrations in the MIMO case is directly related to the fact that smaller antennas will be used, less power will be detected and SNR will be smaller. Cross-coupling may have an additional role but that is definitely a second order effect. I would recommend to review Radar Principles by Toru Sato. https://ntrs.nasa.gov/archive/nasa/casi.ntrs.nasa.gov/19910017301.pdf I would also recommend to review the work of Woodman(1991) which is very related to the type of analysis performed in this work.

4. In the discussion about the baseline cross-correlation, it is not clear why equations 20 and 23 (pages 10 and 11) should give different results. Both expressions come from taking the Fourier transform of a gaussian blow. It seems the difference comes from a different interpretation of the geometry. So, if the same interpretation is given both results (the far field and near field expressions) should be the same. Then, let me ask what the "near field" effects are.

In fact, let me mention the following. In the work of Woodman (1997), it is argued that the near field effect can be modeled as a phase correction in the visibility domain, however, in the present manuscript the near field effect is not presented as a phase correction but as a change of the magnitude of the visibility (correlation) function. Given the different interpretation of the near field effects, I should ask again if the there is actually a "near field" effect that has to be considered in radar imaging problems.

Let me add one more detail. Woodman(1991) derives an expression for the cross-correlation between the voltages of two different antennas showing that the cross-correlation is equal to the Fourier transform of a Brightness function to a second order approximation. In this derivation, there was no need to match the Fraunhofer condition, it was enough that the radar range should be much greater than the separation between the antennas (R»D). This result was actually a generalization of an earlier result presented by Kudeki(1990).

This is a very important issue that needs to be reviewed more carefully in this manuscript. Since it is argued that "near field" effects are considered, the authors

should show clearly what these effects are. However, based on previous literature, it seems that the Fourier transform approximation is good enough for the EISCAT 3D scenario. If that is the case, the problem presented in the manuscript gets simplified and the results presented can be obtained without a complicated framework.

R. D. Palmer, S. Gopalam, and T.-Y. Yu, "Coherent radar imaging using capon's method," Radio Science, vol. 33, pp. 1585–1598, November-December 1998.

T.-Y. Yu, R. D. Palmer, and D. L. Hysell, "A simulation study of coherent radar imaging," Radio Science, vol. 35, pp. 1129–1141, September-October 2000.

J. L. Chau and R. F. Woodman, "Three-dimensional coherent radar imaging at Jicamarca: comparison of different inversion techniques," Journal of Atmospheric and Solar-Terrestrial Physics, vol. 63, no. 2-3, pp. 253–261, 2001.

R. F. Woodman, "A general statistical instrument theory of atmospheric and ionospheric radars," Journal of Geophysical Research, vol. 96, pp. 7911–7928, May 1991.

Kudeki, E., Sürücü, F., and Woodman, R. F. (1990), Mesospheric wind and aspect sensitivity measurements at Jicamarca using radar interferometry and poststatistics steering techniques, Radio Sci., 25( 4), 595– 612, doi:10.1029/RS025i004p00595.
* * *

---

## Author Comment (AC1) · 18 Sep 2020

**Response to anonymous referee #1**

We thank the referee for the careful review and useful comments, which have been helpful to us when improving our manuscript. The reviewer comment is indicated in bold and prefixed with a ">" symbol. Our answers are interleaved with the referee comments below.

>**The paper has some interesting ideas and potential for the incoherent scatter radar (ISR) community. This paper is jumping to apply MIMO radar techniques to ISR and makes a good case that this can be applied to the E-region of the ionosphere. It argues that Eiscatt 3-D will be able to do some interesting high resolution imaging of the E-region ionosphere. The authors simulate the MIMO radar systems and then find that the SVD seemed to give the best results.**

>**The application of generalized inverse techniques to geospace sensors is something I generally support. I think this paper does a good job at taking specific problem in ISR and tries to apply general inverse theory and gets some promising results. Overall the authors did a good job of arguing that this sort of techniques will be useful for ISR. I think they need add some caveats to clarify the applicability of the technique.**

>**First off the the technique the authors show does not include fitting for ISR spectra/ACFs to get plasma parameters. This is fine if they specify that this for the E-region only and that is not clear from the title or the abstract. Yes, different techniques are required for different regions of the ionosphere with ISR but it would help the reader understand the current limits of the method you are applying!**

You are correct in saying that a measurement of the full incoherent scatter radar spectrum is needed in order to fit for the plasma-parameters. However, this paper focuses on characterizing the imaging capabilities of EISCAT 3D, given the current interferometer design. In order to avoid making this study overly complicated. we felt the need to make a simplification of what parameter is to be estimated.

One of the primary purposes of the EISCAT 3D radar imaging will presumably be studying the structure of the ionosphere during auroral precipitation. One of the key parameters in this case is electron density. This needs to be measured with as high temporal resolution as possible. It is to first order proportional to returned power $P = n_e (1+T_e/T_i)^{-1}$ . In the E-region the $T_e/T_i$ ratio is somewhat constant, so we can to first order assume that a radar image of the returned power is related to the spatial distribution of electron density.

Fitting of the ISR spectrum or ACFs is not the main topic of the article. The ACF is indirectly mentioned when calculating the integration time where all time lags are used for analyzing the backscattered power. Implicitly the assumption of E region is repeated here. Clarifying the restriction to the E region we also regard as important. In the manuscript, we changed the beginning of the sentence on p. 5, l. 22-24:
"If we additionally can assume that the autocorrelation function is constant, the number of lagged product measurements per transmit pulse is $N_P (N_P - 1)/2$ because we also can use measurements with different time lags."
to "In the E region, we can assume that the autocorrelation function is constant. Then the number of (...)"

**>I want to also point out there has been work on imaging ISR in the past such as seen in [1], but this was a study that only applied interpolation techniques to ISR data. I think comparing this 2010 paper with the methods here would be helpful and show that this work is new.**
Moving the radar beam to get a wider spatial coverage is somewhat different from the imaging this article discusses. However, this also is a method to see the horizontal distribution of ionospheric parameters, only in an other order of magnitude. For improving the survey over the literature, we move the last part of the paragraph on p.4, l.20-25 to p. 2, where literature on imaging is mentioned.

(The paragraph starts with "The application of aperture synthesis imaging for radar, i.e., aperture synthesis radar imaging (ASRI), has been used in space physics for observing high signal to noise ratio targets (Hysell et al., 2009; Chau et al., 2019)."
Here we will insert
"There is a good amount of literature on ASRI techniques in two dimensions (range and one transverse beam axis direction) for imaging field aligned irregularities (e.g., Hysell and Chau, 2012, and references therein). There have also been several researches on imaging of atmospheric and ionospheric features in three dimensions, eg. Urco et al. (2019), who applied it to observations of PMSE with the Middle atmosphere ALOMAR radar system (MAARSY), Palmer et al. (1998) on data from the middle and upper atmosphere radar in Japan, Yu et al. (2000) with a simulation study, and Chau and Woodman (2001) on the atmosphere over Jicamarca."
After this comes "The currently available horizontal resolution [insert "of ASRI"] is around 0.5° with Jicamarca, but down to 0.1° for strong backscatter (Hysell and Chau, 2012) in the case of field-aligned ionospheric irregularities; and 0.6° with MAARSY for polar mesospheric summer echoes (PMSE) (Urco et al., 2019)."

The next sentences should be a new paragraph:
"However directly on incoherent scatter in three dimensions there is little literature, but some approaches have been made, like Schlatter et al. (2015), who used the EISCAT Aperture Synthesis Imaging array and the EISCAT Svalbard radar to image the horizontal structure of Naturally Enhanced Ion Acoustic Lines (NEIALs) and Semeter et al (2008) who interpolated sparse multio-beam PFISR-measurements to estimate the E-region electron density variation over a 65x60 km area during an auroral event.")

**>The following is a list a technical corrections I've found will need to fixed:**

**>- Page 6 line 25: Do mean truncated cone? Conic section implies a 2-D plane and this is a 3-D volume.**
Yes, we mean a truncated cone. The other referee recommended to switch to a spherical cone as this figure represents the radar volume better and this will be included in the revised manuscript.

**>- Page 7 line 1: You state range, which range is this because we're not in a multistatic set up and the term range is ambiguous, transmit, receive, bistatic?**
The imaging interferometer antennas for EISCAT 3D can be considered as approximately a monostatic system, because the largest separation between antennas is less than 2 km. In this case, the transmitter-target-receiver range is nearly constant for all antennas.

We have added the following  statement in the paper to hopefully make this issue more clear to the reader: "by range, we mean the range from the center of the core array in Skibotn to the target." after mentioning the range for the first time on p. 7, l. 2.

**>- Page 7 lines 9-10: O2 + is used for the calculation but the dominate species in the E-region is NO+. I don't think this will create a huge error in the calculations but it should be mentioned that this will not throw off the calculation by much.**

You are right. Fortunately, the molecular masses are very similar. According to Brekke (2013), p. 222, the dominating ion species in the E region are $NO^+$ and $O_2^+$ where there is slightly more of $O_2^+$ around 120 km, else $NO^+$ dominates, but the difference is small. Since the species are approximately equally common, we can change the ion mass to 31 u corresponding to a mixture of $NO^+$ and $O_2^+$.

In the article, the ion mass is only used for calculating the thermal velocity, which is used for calculating the bandwidth of noise. The noise bandwidth is chosen by taking the largest of the thermal velocity and Since the inverse of the bit length exceeds twice the thermal velocity times the wave number by at least one order of magnitude, changing the ion mass slightly has no effect on the signal-to-noise-ratio.

In the paper, we will change "where $m_i$ is the ion mass, which we set equal to 32 u corresponding to $O_2^+$" to "(…) to 31 u corresponding to a mixture of $O_2^+$ and $NO^+$. These are the two most dominant ion species in the E region (Brekke 2013)." (l. 9). At the end of the paragraph, we will add "For all bit lengths investigated here, $\tau_b^{-1}$ exceeds $2v_{th}k$ by at least one order of magnitude. The bandwidth is therefore independent on the ion composition as long as the measurements are restricted to the E region."

**>- Page 8 Figure 2: I'm having trouble seeing the need for this figure since the assumption is for a monostatic system, and chemical composition is off too. I'm aware with MIMO that the resolution can be chosen but this link budget only accounts for a monostatic system. Plus in a monostatic system the gain from the antenna is directly tied to the resolution.**

We feel this figure is of primary importance. It tells us if imaging is feasible or not from the point of view of signal to noise ratio. The figure shows the expected time resolution achievable with EISCAT 3D.

The caption for Figure 2 states the following: "Integration time of targets in the E-region observed using the E3D core for transmit and a single 91 antenna element module for receive".

A single 91 element antenna module is the antenna module used for interferometry. The integration time is the required integration time to estimate a cross-correlation function between antenna modules, which is the basic measurement that goes into imaging.

The gain of an antenna module is significantly less than it is for the full core array receiver. We therefore felt it to be important to investigate if it is even possible to make a measurement of incoherent scatter radar with reduced gain on the receiver.

Knowing how long it takes to estimate a cross-correlation between interferometer antennas with a certain accuracy is of primary importance when determining how long it takes to make a radar image with this system.

**>- Page 13 lines 13-14: Please be more clear about what was done to the image from Figure 1. It seems you just took the magnitude of the image and mapped it from 0 to 1?**

Yes, this is correct. We will add a sentence on this to p. 18, l. 14 where we mention the image: "(As original image, we use a part a part of Fig.1). A part of 97 x 97 pixels was cut out of the figure and the greyscale values were mapped to lay between 0 and 1. (From the...)"

**>Overall, good paper. I think it should be accepted with the changes I've go over above.**

**>[1] J. Semeter, T. Butler, C. Heinselman, M. Nicolls, J. Kelly, and D. Hampton, "Volumetric imaging of the auroral ionosphere: Initial results from PFISR," J. Atmos. Solar-Terrestrial Phys., vol. 71, no. 6–7, pp. 738–743, May 2009.**

Thank you for your comments.

---

## Author Comment (AC2) · 18 Sep 2020

**Response to anonymous referee #2**

We thank the referee for the throughout review which has helped us to improve the manuscript. The review is repeated here in bold and starting with an arrow. Our comments are written below.

>**The present manuscript analyzes the problem of radar imaging in 3D for incoherent scatter applications that will be implemented using the EISCAT 3D radar. It is mentioned that the proposed technique includes "near field" effects on the formulation of the radar imaging problem because EISCAT 3D applications will be in such regime. The analysis includes also the concept of MIMO radars in order to improve the resolution of radar images. The manuscript is well organized and the results are presented clearly. Although the analysis performed in this document introduces new ideas related to the radar imaging problem, I would recommend a careful revision of the document before its possible publication. As I will explain there are some important issues that have to be addressed first.**

>**1. In Equation 2 (page 5), it is assumed that the number of independent measurements per second is proportional to the number of lagged products in a longpulse experiment. This is definitely not the case. In a long pulse experiment, lag products are not independent, all of them are correlated. This is because, within the length of a pulse, signals from a common volume are mixed. Assuming that all lag products are equally informative, it is also an oversimplification that I think may lead to not necessarily correct conclusions, particularly, in this application in which the target fills the radar volumen. I would recommend the authors to review this section in order to analyze more carefully the relationship between the number of samples within a longpulse and the integration time needed to reduce statistical uncertainty. Notice that if you consider Np=1, there is a singularity in equation (7), I don't think this is correct. I would also recommend to review equation 11 since a radar volume can be modeled better as a spherical cone section rather than as a truncated cone. In this expression, if you consider "r" at the center of the radar volume, the expression becomes simplified.**

In the lagged products, the signal is correlated, but white noise is not. As long as the noise power is much greater than the signal power, also clutter and other non-white noise effects can be neglected. The lagged products are therefore independent for low SNR. At zero lag, the product includes all the white noise from the receivers. We therefore ignore the zero lag. This is where the singularity in eq. (7) comes from. If including the zero lag, the denominator would be $F_m N_p (N_p+1)$ without singularity. When inserting Np = 1 into Eq. (7), and the zero lag is ignored, there are no measurements left and the variance is infinite.

In the E region, the decorrelation time is long in the VHF band which is due to heavy ions ($O_2^+$ and $NO^+$) and relatively low electron and ion temperature. While the pulse is 0.5 ms long, the decorrelation time is around 1 s.

We have investigated the difference between a truncated cone (conical frustum), a spherical cone section, and a cylinder when modeling the volume in the E-region using a radar beam corresponding to the solid angle of the EISCAT 3D beam. We found no significant differences between these three models in this case. This study is included in the referee response.

The radar volume in Eq. (11) is indeed better represented as a spherical cone than by a truncated cone. Changing the model to a spherical cone has the consequence that $\tan^2(\theta/2)$ is substituted with $2(1-\cos(\theta/2))$. For small $\theta$, like $\theta = 1°$ as in the article, the difference is small. For significantly larger beam opening angles (more than 10 degrees), these models start to diverge. Considering r to be the range to the center of the radar volume simplifies the expression in the brackets to $3r^2+\Delta r^2/4$

and the volume shrinks about 1%. The equation in the manuscript will be changed to the spherical cone, but letting the range be to the lower boundary of the volume as before since this is closer to what was used in the calculations.

**>2. In the introduction (line 22, page 4), it is mentioned that there is not much literature related to 3D imaging and the authors make reference to a recent work of one of the coauthors. This is not fully true, the works of Palmer et al(1998), Yu et al (2000), and Chau & Woodman (2001) (see references below) addressed the imaging problem in 3D in the same sense as the present manuscript does. Of course the difference is that the new approach is addressing the incoherent scatter problem while the previous work was mainly focussed on coherent scatter echoes. So proper references should be used.**
We will add the references mentioned. Since the literature description in the article starts to become complex because similar literature is described two places in the text, we merge the literature on imaging on p.2. Here we will clarify that the novelty is the use on incoherent scatter.

**>3. In line 29, page 7, the integration time for MIMO applications is analyzed and it is mentioned that the integration time will be longer in the MIMO case than in the SIMO case, but the authors indicate that the difference depends on cross-coupling between antennas. I don't think this conclusion is correct, at least not as a first approximation. There is plenty of literature related to soft-target radar equations that explain clearly that the received power is directly proportional to an effective antenna aperture area (which is also proportional to the true antena area). So, even if you use the same power on transmission, the received power will be less when using a small antenna. Then, the need for additional integrations in the MIMO case is directly related to the fact that smaller antennas will be used, less power will be detected and SNR will be smaller. Cross-coupling may have an additional role but that is definitely a second order effect. I would recommend to review Radar Principles by Toru Sato. https://ntrs.nasa.gov/archive/nasa/casi.ntrs.nasa.gov/19910017301.pdf I would also recommend to review the work of Woodman(1991) which is very related to the type of analysis performed in this work.**

We were unaware of the Toru Sato and Woodman papers. We have studied them and they both seem like useful references on how atmospheric radars operate.

When using MIMO, the transmit antenna is divided into N separate transmit sections. These different regions need to transmit different waveforms in order for us to be able to separate the different transmitter sections on receive.

If we divide the antenna into N parts (and thus N independent transmitters), each transmit section will have an aperture of A/N and a power of P/N. Here A is the total area of the full array and P is the total power of the full array. This is the ideal case.

In discussions with EISCAT staff, we have been told that two neighbouring regions of the antenna should not be transmitting simultaneously with different codes, as the mutual coupling of two different transmit signals might be problematic. This mutual coupling may in the worst case cause power amplifiers to overload and break.

It was suggested that in order to reduce mutual coupling of different regions of the antenna when dividing it into multiple transmitters, buffer zones could be made around each section of the antenna array. This would reduce the amount of area and power for each section, making it slightly less than A/N and P/N.

We have tried to carefully reword this in our manuscript to make this point more clear.
Now the last part of the paragraph says:
"The transmit gain must be divided by the number of transmitters. It could be that because of cross-coupling between antennas, there must be buffer zones between transmitters. Then the gain decreases furthermore. On the other hand, the radar will illuminate a larger volume that contains more scatterers and so increase the received power again. In conclusion, the integration time for MIMO will be longer than for SIMO. How long is mostly dependent on the possible cross-coupling between antennas."
We will change it to:
"Because of the smaller antenna area, also the transmit gain must be divided by the number of transmitters. Additionally, there could be cross-coupling between antennas, which force buffer zones between transmitters. Then the antenna area and gain decrease furthermore. In conclusion, the integration time for MIMO will at least be the number of transmitters times the integration time for SIMO."

**>4. In the discussion about the baseline cross-correlation, it is not clear why equations 20 and 23 (pages 10 and 11) should give different results. Both expressions come from taking the Fourier transform of a gaussian blow. It seems the difference comes from a different interpretation of the geometry. So, if the same interpretation is given both results (the far field and near field expressions) should be the same.**
Equation (23) truly represents the farfield, and Eq. (20) was derivated mostly in nearfield. However to be integrateable, between Eq. (16) and (17) there was done an approximation to make the exponential linear. The approximation can be interpreted as assuming plane waves. Therefore, Eq. (20) is not exact anymore.

**> Then, let me ask what the "near field" effects are.**
The nearfield effects are blurring of the image as can be seen in the image reconstructions with matched filter. Palmer et al. (1998) call the method "Fourier-based imaging" because it uses the Fourier transform for reconstruction, which implies the farfield approximation.

**>In fact, let me mention the following. In the work of Woodman (1997), it is argued that the near field effect can be modeled as a phase correction in the visibility domain, however, in the present manuscript the near field effect is not presented as a phase correction but as a change of the magnitude of the visibility (correlation) function. Given the different interpretation of the near field effects, I should ask again if the there is actually a "near field" effect that has to be considered in radar imaging problems.**
The phase correction in Woodman (1997) can probably be used for imaging with EISCAT 3D. In the study we however followed an other approach where we do the simulations completely in the nearfield. As we say in the introduction, the computation becomes more complex, but is accessible with modern computers.

**>Let me add one more detail. Woodman(1991) derives an expression for the cross-correlation between the voltages of two different antennas showing that the cross-correlation is equal to the Fourier transform of a Brightness function to a second order approximation. In this derivation, there was no need to match the Fraunhofer condition, it was enough that the radar range should be much greater than the separation between the antennas (R»D). This result was actually a generalization of an earlier result presented by Kudeki(1990).**

**>This is a very important issue that needs to be reviewed more carefully in this manuscript. Since it is argued that "near field" effects are considered, the authors should show clearly what these effects are. However, based on previous literature, it seems that the Fourier transform approximation is good enough for the EISCAT 3D scenario. If that is the case, the**

**problem presented in the manuscript gets simplified and the results presented can be obtained without a complicated framework.**

It seems that Woodman (1991) assumes plane waves in a similar form as the linearization mentioned above. With the convention Toru Sato refers to, everything closer than ~1000 km is in the nearfield, if including the EISCAT 3D outrigger subarrays. The Fourier transform with correction as described by Woodman (1997) might be good enough for EISCAT 3D, but it is possible to calculate the theory matrices and do the simulations in the nearfield taking into account the spherical nature of the backscattered wavefronts and the antenna geometry. In general, when solving inverse problems accurate theory matrices are important.

In practice, when we have some imaging measurements from EISCAT 3D and we want to reconstruct the image with SVD, only the theory matrix A is needed. Regardless of near- or farfield, the SVD itself requires the most computational power. However, after having been computed once, it can be saved and reused.

**>R. D. Palmer, S. Gopalam, and T.-Y. Yu, "Coherent radar imaging using capon's method," Radio Science, vol. 33, pp. 1585–1598, November-December 1998.**

**>T.-Y. Yu, R. D. Palmer, and D. L. Hysell, "A simulation study of coherent radar imaging," Radio Science, vol. 35, pp. 1129–1141, September-October 2000.**

**>J. L. Chau and R. F. Woodman, "Three-dimensional coherent radar imaging at Jicamarca: comparison of different inversion techniques," Journal of Atmospheric and Solar-Terrestrial Physics, vol. 63, no. 2-3, pp. 253–261, 2001.**

**>R. F. Woodman, "A general statistical instrument theory of atmospheric and ionospheric radars," Journal of Geophysical Research, vol. 96, pp. 7911–7928, May 1991.**

**>Kudeki, E., Sürücü, F., and Woodman, R. F. (1990), Mesospheric wind and aspect sensitivity measurements at Jicamarca using radar interferometry and poststatistics steering techniques, Radio Sci., 25( 4), 595– 612, doi:10.1029/RS025i004p00595.**

Thank you for your comments.

Brekke, Asgeir (2013): "Physics of the upper polar atmosphere", Springer, Heidelberg